

# Assessment of the observability of coastal currents in LRM and SAR altimetry observations: a north-western Mediterranean Sea case study

**Alice Carret[1,2], Florence Birol[1], Claude Estournel[1], Bruno Zakardjian[3]**

[1]LEGOS,Université de Toulouse-CNES-CNRS-IRD, OMP, 14 Av. E. Belin, 31400 Toulouse, France

[2]SERCO, Via Sciadonna 24-26, Frascati, Rome, Italy

[3]Université de Toulon, CNRS/INSU, IRD, Mediterranean Institute of Oceanography (MIO), UM 110, 83957 La Garde, France

Correspondence to: Alice Carret : alice.carret@legos.obs-mip.fr

## Abstract

Over the last three decades, satellite altimetry has observed Sea Surface Height variations, providing a regular monitoring of the ocean circulation. Altimetry measurements have an
intrinsic signal-to-noise ratio that strongly limits the space scales, and then the geophysical information that can be captured with this instrument. However, the recent progress made on both altimetry sensors and data processing, offers new perspectives in terms of research-oriented and operational applications. In this paper we present a methodological study that helps to better quantify the impact of this progress in terms of coastal circulation.

We focus on a case study: the Northern Current, a narrow slope current (less than 60 km wide) located in the North Western Mediterranean Sea. We first use a high resolution numerical model validated with HF radars and underwater glider data to define the general characteristics of the Northern Current in terms of surface velocity and sea surface height signature. These characteristics are then compared with corresponding estimates of sea
surface height velocity derived from 1-Hz altimetry data sets from three missions: Jason 2 (Ku-band LRM), SARAL (Ka-band LRM) and Sentinel-3A (SAR). The data from all missions were processed with the coastal-specific X-TRACK strategy.

We show that near Toulon, the model is very close to the observations in terms of current estimates, providing a very good reference for altimetry data located in this area. The
Northern Current is observed 15 km to the coast on average, with a mean core velocity of 0.44 m s[-1]. Its signature in sea level consists of a drop whose mean value at 6.14°E is 6.9 +/- 2.2 cm extending over 18 +/- 4 km. These variations show a clear seasonal pattern, but high frequency signals are also present most of the time. In 1-Hz altimetry data, the mean sea level


drop associated with the Northern Current is overestimated by 3.6 cm for Jason 2, 0.3 cm for
SARAL and 1.4 cm for Sentinel-3A. In terms of corresponding sea level variability, Jason 2
and SARAL altimetry estimates are larger than the model reference (+1.3 cm and +1 cm,
respectively) whereas Sentinel-3A shows closer values (-0.4 cm). Without any sea level data
filtering, the standard deviation of altimetry-derived velocity values is 3.7, 2.4 and 2.9 times
too large for Jason-2, SARAL and Sentinel-3A, respectively. When filtering sea level data,
the distribution of altimetry velocities tends to converge towards the model reference with a
50-km, 30-km and 40-km cutoff wavelength for Jason-2, SARAL and Sentinel-3A data,
respectively.

# 1. Introduction

Since the beginning of the 90s, satellite altimetry has enabled many regional circulation
studies (e.g. Troupin et al., 2015; Vignudelli et al., 2000 in the NW Mediterranean Sea;
Gourdeau et al., 2017 in the Solomon Sea; Liu et al., 2018 in the South China Sea, …). Its
main advantages are its long-term and regular temporal coverage and its synoptic character.
Large scale structures (>150 km) are well captured with this observational technique which
has a crucial role in the knowledge of the circulation at global scale (Fu and Le Traon, 2006).
On the contrary, meso-scale and sub-meso-scale processes such as eddies and meanders or
narrow coastal currents are historically poorly resolved by altimetry and generally
documented by in situ observations or numerical models (e.g. for the NW Mediterranean Sea:
Casella et al., 2011; Guihou et al. 2013; Juza et al., 2013; Ourmières et al., 2010; Schroeder et
al., 2011). However, during past years, new altimetry techniques such as the Ka-band
frequency and the SAR (Synthetic Aperture Radar) mode, as well as progress in data
treatment, have led to a significantly gain in observability of fine scale ocean structures in
general and of coastal features in particular (Birol et al., 2021; Morrow et al., 2017; Verron et
al., 2018).

Despite the progress made, intercomparisons with in situ observations of near-coastal
currents have shown that the corresponding altimetry-derived surface velocities are
underestimated (Birol et al., 2010; Jebri et al., 2016). In Carret et al. (2019), using long time
series of both ADCP and glider data as a reference for the Northern Current (NC hereinafter)
velocities, we have shown that satellite altimetry data underestimate the amplitude of NC
seasonal variations by ~40-45 %. This can be explained by the ageostrophic current
component, not captured by altimetry, but also by the effective data resolution, which is
limited by the altimeter noise and coastal data processing issues, resulting in near-shore data
gaps. The impact on signal observability varies from one altimeter instrument to another,
decreasing with new radar techniques and data processing approaches (Birol et al., 2021;
Morrow et al., 2017). Nevertheless, there is still a need to specify more precisely the
improvements in coastal observability related to differences in altimetry technologies (Ka-



band, SAR altimetry) and processing techniques. In order to optimize the use of altimetry in near-shore areas and to finally define its place among other coastal observation systems, it is
important to better study the dynamic processes that can be observed and to quantify the associated accuracy.

As satellite altimetry measures sea surface height (SSH or sea level hereinafter), the
observability condition is that the processes of interest have a sea level signature and spatio-temporal scales larger than the altimetry resolution. Over the open ocean, the altimetric observability problem is generally studied through a spectral approach (Dufau et al., 2016; Morrow et al., 2017; Vergara et al. 2019). This gives a mean statistical solution over the considered region, but can not be used in the coastal ocean where too short satellite track
sections often impede the computation of a spatial spectral analysis. Several studies (Bouffard et al., 2008; Carret et al., 2019; Pascual et al., 2015; Troupin et al., 2015) have used in situ observations to analyze the resolution capability of coastal altimetry data but they came up against the scarcity of independent measurements and their non-colocation in space and/or time.

In this paper, we propose a different strategy based on a high resolution numerical model. Our purpose is to quantitatively analyze if a particular coastal dynamical structure can be observed with satellite altimetry. Using a high resolution model may help to overcome the issue of colocation between in situ and altimetry, but given the essential condition that the physical process studied must be correctly represented by the model. After a careful model
validation step in the study region, our approach will consist in using the model to quantify the SSH signature of an identified physical process along a particular satellite track. In a second time, the3 model solution will be considered as the reference and compared with the SSH signature captured in the altimetry dataset along the considered tracks. The observed differences will be analyzed and quantified through different diagnostics.


The case study chosen is again the NC in the Western Mediterranean Sea (NWMed hereinafter). This region is considered as a laboratory area for coastal altimetry studies (Birol et al., 2010; Birol and Delebecque, 2014; Bouffard et al., 2008) because of its small Rossby
radius (around 10 km, Grilli and Pinardi, 1998) leading to a wide variety of mesoscale and submesoscale structures. We can also benefit from the variety of in situ data collected from the MOOSE (Mediterranean Ocean Observing System for the Environment, https://www.moose-network.fr/) integrated observing system and of the long experience and really good performances previously obtained with the high resolution SYMPHONIE
numerical model in the study area (Damien et al., 2017; Estournel et al., 2016; Herrmann et al., 2008).

The NC is a narrow slope current (Fig. 1) formed by the junction of the Eastern (ECC) and



Western (WCC) Corsica Currents in the Ligurian Sea (Taupier-Letage and Millot, 1986). It
flows cyclonically along the Italian, French and Spanish coasts (Millot, 1987). It has a strong
seasonal component with a maximal transport (1.5-2 Sv, Alberola et al., 1995) and increased
mesoscale variability in winter (e.g. Crépon et al. 1982; Flexas et al. 2002; Sammari et al.
1995). Its position relative to the coast also varies through the year, from less than 20 km to
about 30 km from the coast (Niewiadomska, 2008; Sammari et al., 1995), as well as its depth
(more than 200 m in winter and 150-200 m during the rest of the year) and width (30 km with
a narrowing in winter, Alberola et al., 1995).

In the past, the NC variability has been intensively studied with in situ observations and
models: mesoscale fluctuations in Sammari et al. (1995) ; eddies associated in Casella et al.
(2011), Hu et al. (2011) and Schaeffer et al. (2011). Birol et al. (2010) have highlighted the
contribution of along-track satellite altimetry to study the NC variability. Since then, other
studies have used such data to investigate the NC circulation as well as the recirculation and
associated meanders (Borrionne et al., 2019; Morrow et al., 2017; Pascual et al., 2015).

Here, we will investigate in details the NC observability issue for three altimetry missions
associated to different techniques: Jason 2, with the classical Ku-band LRM nadir altimeter,
SARAL which uses the Ka-band frequency in LRM mode and Sentinel-3A (Sentinel-3
hereinafter) with its Synthetic Aperture Radar mode. Section 2 describes the study tools and
the model validation step. Section 3 presents the methodology used to quantify the NC sea
level signature in the Ligurian Sea and the results obtained. Section 4 focuses on the NC
observation with the three altimetry missions and analyzes the differences obtained between
altimetry and the model. Section 5 summarizes and concludes the paper.

# 2. Study tools

In this study, in situ (glider) and HF radar data are first used to validate a regional numerical
simulation (section 2.3). Our study period, strongly constrained by both the in situ data and
model simulation availability, goes from 2011 to 2019.

## 2.1. Data

### 2.1.a) HF radars

We took advantage of the 2 years of data, from May 2012 to September 2014, provided by
the HF Wellen radar (WERA) instruments installed near Toulon as part of the MOOSE
network (Zakardjian and Quentin, 2018). It corresponds to the dataset available at the time of
the study. The stations are located in Cap Sicié and Cap Bénat-Porquerolles in respectively
monostatic and bistatic eight-antenna configurations (now upgraded to twelve antenas by site,



Dumas et al., 2020). Their positions enable to monitor the NC upstream the Gulf of Lions (Fig. 2a) and the mesoscale dynamics that occur in this region of cross-shelf exchanges and strong atmospheric forcing (Mistral, Tramontane winds). They operate at 16 MHz with a 50 kHz bandwidth, resulting in a spatial resolution of 3 km, and allow an angular resolution of 2°. The radars provide the surface current every hour over a region of 60 x 40 km. Data are then filtered from tides and inertial oscillations, edited and averaged daily and finally bined on a regular 2x2 km grid.

### 2.1.b) Gliders

In the NWMed, a number of gliders have been deployed since 2005 along different transects, measuring temperature and salinity vertical profiles. We focus on a regular line, from Nice to Calvi where 36 deployments occurred from 2009 to 2016, as part of the MOOSE network. From 2011 to 2017, there are 204 sections (ascending or descending). Data were treated according to Carret et al. (2019) who discarded profiles being too short or deviating too much from an average Nice-Calvi trajectory. It results in temperature and salinity data down to 500m (depth reached by all gliders), gridded with a 4 km horizontal bin size along the mean trajectory considered as a reference track. The temperature and salinity data are then filtered using a 15 km cutoff wavelength. The geostrophic velocity component perpendicular to the reference track is then derived using the thermal wind equation referenced to 500 m (see Carret et al., 2019 for further details).

### 2.1.c) Satellite altimetry

Jason 2 was launched in June 2008 and was in the same orbit up to October 2016. It is based on a conventional Low Resolution Mode (LRM) altimeter operating in the Ku-band and has a 10-day repetition cycle. SARAL, launched in February 2013 moved to a drifting orbit in July 2016, providing a shorter data time series (3 years). It has a 35-day repeat observation cycle. However, with its LRM altimeter operating in the Ka-band (called AltiKa), it has a lower data noise and is expected to capture smaller spatial scales than Jason 2 (Verron et al., 2018). Sentinel-3 was launched in February 2016. With its SAR (Synthetic Aperture Radar) altimeter, its footprint is further reduced in the along-track direction, compared to SARAL. Sentinel-3 has a 27-day repeat observation cycle. Figures 2b,c,d indicate the satellite tracks of each mission in the NWMed, defining the spatial coverage of the corresponding nadir altimetry observations. Note that the spatial resolution of nadir 1-Hz altimetry data is in the range 5-8 km along the track (Table 1) but that the inter-track distance varies from 230 km for Jason 2 to 76 km for Sentinel-3 and 58 km for SARAL. For each mission, the tracks used in this study are indicated in bold in Fig. 2b,c,d. They correspond to the tracks closest to HF radars data (see below for explanation): the Sentinel-3 track 472 and the SARAL track 302 pass over the HF radars region with a different angle, whereas the Jason 2 track 222 is located a bit further to the east, at about 60 km. Table 1 summarizes the characteristics of each altimetry dataset.


For all missions, we use the X-TRACK regional product processed with a coastal oriented strategy described in Birol et al. (2017) (DOI: 10.6096/CTOH_X-TRACK_2017_02) that provides 1-Hz Sea Level Anomaly (SLA) time series homogeneously processed and regularly spaced (Table 1, along-track resolution) along the different satellite tracks. The processing is the same for all missions, except that the dual-frequency of Jason 2 and Sentinel-3 altimeters allows to compute the ionosphere correction whereas a model is required for SARAL. This correction being associated with long wavelengths, it should not impact the results obtained in this study.

From SLA data, the across-track geostrophic current (*u*) can be inferred through the geostrophic equation (Eq. 1) after adding a regional Mean Dynamic Topography (MDT, Rio et al., 2014) .

$$u = \frac{-g}{f} \quad \frac{\Delta(SLA + MDT)}{\Delta x} \qquad (1)$$

where *g* is the gravitational constant, *f* the Coriolis parameter and *Δx* the distance between the 1-Hz altimetry points. Before computing current estimates, the SLA may be filtered in the along track direction in order to remove the remaining altimetry noise. In this study the filtering is applied in Section 4.2 but not in Section 4.1. When used, the filtering is done with a low-pass Loess filter using different cut-off wavelengths (see Section 4.2) .

## 2.2. Model

We rely here on the SYMPHONIE primitive equation model which has been widely used in the study area at the nearshore (Michaud et al., 2012), coastal (Estournel et al., 2003; Mikolajczak et al., 2020; Petrenko et al., 2008) and regional (Estournel et al., 2016) scales.

SYMPHONIE is described in Marsaleix et al. (2008, 2006), Damien et al. (2017), with turbulence closure and convection parameterization detailed in Estournel et al. (2016). The configuration used in this study covers the whole Mediterranean basin, the Marmara Sea and extends westward up to 8°W in the Gulf of Cadiz; it is described in Estournel et al. (2021). The horizontal resolution is minimum (2 km) in the northwestern Mediterranean (except for a local narrowing at the Gibraltar strait). A VQS (vanishing quasi-sigma) vertical coordinate (Estournel et al., 2021) with 50 levels is used. The model is initialized and forced at its open boundaries with analysis produced by the operational oceanography center MERCATOR OCEAN International, (MOI, Lellouche et al., 2013). As stratification is crucial for mesoscale characteristics, it has been debiased from observations collected over the whole basin as in Estournel et al. (2016) while preserving the first hundred meters which benefits optimally from the data assimilation performed at MOI. At the air/sea interface the hourly forecasts of ECMWF at the horizontal resolution of 0.125° are used to calculate heat and momentum fluxes through bulk formulae.



The model simulation covers the period from 18 May 2011 to 31 March 2017 and provides 4-day averaged fields. Daily outputs are also available during the period of HF radars availability and will be used in the validation exercise (section 2.3).

## 2.3. Simulation validation

The first step of this study is to validate the model simulation in terms of surface current which is here the variable of interest. The currents deduced from the gliders and HF radars are compared to their equivalent in the simulation: geostrophic ones for the glider, total currents for the radars. The representation of the NC is compared using statistics (time-average and standard deviation) and time-space diagrams (Hovmuller diagrams).


For the comparison with the HF radars, we consider the zonal current component from May 2012 to September 2014 along a section located at 6.14°E, just south of Toulon (Fig. 2). Note that, due to the coast configuration in this area, the NC which follows the 1000-2000m isobaths is mainly westward, i.e. with a dominant zonal component most of the time (with the exception of short living, 3-6 days, meanders or wind-induced instabilities). Figure 3a shows
the time-average and standard deviation of the zonal velocity as a function of latitude along this section. At this longitude, the NC flows westward and corresponds then to the negative values observed north of 42.7°N. In terms of statistics, there is an excellent agreement between the HF radars and the simulation. On average, the NC position and current amplitude
are almost identical in both fields. The maximum NC amplitude (called Vmax hereinafter) is $-0.44 \pm 0.16$ m s$^{-1}$ for the simulation and $-0.43 \pm 0.19$ m s$^{-1}$ for the radars. This velocity value, identified as the NC core, is located at 42.85°N for both simulation and observations. We define the width of the NC as the length of the section around its core where the absolute velocity is larger than |Vmax|/2. On average, it is $18 \pm 5.9$ km for the simulation and $18 \pm 6.1$
km for the observations. The main difference along the section is that between the NC and the coast (to the north), where the velocity variability is slightly greater for the HF radars than for the simulation.

In order to investigate the representation of the NC variability in the simulation in more
detail, Fig. 3c represents the time space diagrams of the zonal velocity along 6.14°E for both the HF radars and the simulation and the differences between both fields. We observe an overall good agreement between the observations and the simulation, both estimates showing the same seasonal variability, i.e. larger velocities in winter and spring and a summer slow down, and a similar high frequency variability that may instantiate the wind-induced (Ekman
current) and mesoscale (meanders and eddies) variability of the circulation. The differences between the currents' estimates are generally low and higher values (order of a few tens of cm s$^{-1}$) can be largely explained given the fact that short-living structures may not strictly coincide in time and space in the model and observations.



The same diagnostics have been computed for the simulation and the glider data along the
Nice-Calvi section, located further east (Fig. 3b,d) but in this case with the geostrophic
current component normal to the section. We also observe a good agreement between the
simulation and the gliders but with more differences than what was obtained with the radars,
especially in terms of current variability. We obtain Vmax values of -0.23 ± 0.12 m s$^{-1}$ for the
model and -0.25 ± 0.13 m s$^{-1}$ for the gliders. Near the coast the differences between the
observed and simulated mean currents can reach 0.1 m s$^{-1}$. The NC core is located at 43.51°N
for the simulation and at 43.52°N for the observations. The NC is thus well located in the
simulation, but narrower (24 ± 6.6 km), compared to the observations (30 ± 9.6 km).
Concerning the time-space diagrams, the instantaneous differences in velocity between the
observations and the simulation can reach 0.5 m s$^{-1}$. They are associated with a misplaced
current in the model rather than with incorrect current values. The irregular temporal
sampling of the gliders also contributes to these larger model-data differences, compared to
the HF radars results. Indeed, a deeper analysis shows that the same features may occur in the
simulation and in the observations, but shifted by one or two days (not shown).


Finally, we have used all the observations available (glider, HF radars and altimetry) in order
to have a general view of the model ability to represent the regional circulation in our study
area (Fig. 4). For the sake of clarity, we chose to represent for SARAL only the tracks closest
to in situ observations in order to not overload the figure. We compute the time-average and
the standard deviation of surface current perpendicular to the tracks derived from all
observations and from the simulation collocated in space and time with the observations (Fig.
4). These statistics are computed over the common period of data availability (if we consider
only SARAL and Jason-2 in terms of altimetry missions): from March 2013 to October 2014.
Both a regional view of the NW Mediterranean Sea and a zoomed-in view of the northern
Ligurian Sea are provided. Note that the observation component of these diagnostics (top
panels) have already been shown in Carret et al. (2019) to analyze how the NC is captured by
the different types of instruments while the focus here is on the simulation quality. In Fig. 4a
and c, the NC corresponds to the negative velocity values (westward flow) represented in
blue along the French and northern Spanish continental slope. As already indicated in Carret
et al. (2019), the continuity of the circulation emerges when putting together the different
instruments which show a very good consistency in terms of mean current. The model also
shows this continuity with almost identical current values: around -0.19 m s$^{-1}$ south of Nice
against -0.24 m s$^{-1}$ for the gliders and around -0.42 m s$^{-1}$ south of Toulon against -0.41 m s$^{-1}$
for the HF radars. The standard deviation of the current is larger along the NC. It is captured
by all types of observations but with differences from one instrument to the other (see Carret
et al., 2019 for analysis). These differences due to the sampling bias and the spatial variability
are also observed in the simulated field, although the corresponding standard deviations are
weaker. At the HF radars location, the NC variability results in standard deviation values of
0.24 m s$^{-1}$ for the observations and 0.14 m s$^{-1}$ for the simulation, which is coherent with what
is observed on Fig. 3. The variability is less important for altimetry data which don't get as
close to the coast as in situ observations and measure a different geophysical content than the




observations. Carret et al. (2019) showed that these differences mainly come from the temporal resolution and the number of data samples.

All these results show that the simulation has excellent skills in terms of circulation, as well at the regional scale all along the NC paths than at the local one in the vicinity of the HF radars and glider covered areas.

## 320 3. Signature of the NC on sea level

The good results obtained above in the Ligurian Sea in terms of model-data comparison allow us to use the simulation as a reference for altimetry data analysis. It is first used to quantify the NC sea level signature before analyzing how it is captured by altimetry data (section 4). We first describe how we quantify this signature over the area covered by the HF radars 325 observations.

In the simulation, we first extracted the sea level profiles for each date along the section located at 6.14°E (see Fig. 2a). The corresponding cross-transect surface geostrophic current component is then calculated using Eq. 1, as for classical altimetry estimates.

For each SSH profile, we use three diagnostics to characterize the NC sea level signature. 330 First, the location of the NC core, corresponding to the maximum velocity in absolute value, is spotted on the cross-shore current profile (expressed as a distance to the coast). Then, the drop in SSH (called *diff*) is computed over the region delimited by velocity values higher than half of the NC core velocity (Eq. 2).

$$diff = max\left(SSH_{|u| \geq \frac{|u|max}{2}}\right) - min\left(SSH_{|u| \geq \frac{|u|max}{2}}\right) \quad (2)$$

Finally the width (*dx*) of this region, defining the NC width, is derived as the distance between the two half NC core velocities.

Figure 5 illustrates the methodology described above for the model SSH and corresponding zonal current profiles along the 6.14°E transect and averaged over the HF radars period. The profiles are represented as a function of the distance to the coast. On Fig. 5a, the dashed 340 vertical lines delimit the NC width. They are transposed on Fig. 5b in order to derive the corresponding SSH drop (*diff* value).

We observe that, on average, the SSH decreases from 8 km to 28 km to the coast, i.e. the distance *dx*. This corresponds to the NC associated with negative zonal velocity values. Still on average, the NC core velocity is -0.39 m s$^{-1}$ and is at about 18 km from the coast. It 345 corresponds to a drop in sea level of 6.9 cm over 20 km. These values are considered as the mean sea level signature of the NC in the area considered.



The time series of the three diagnostics defined above along the 6.14°E transect are represented in Fig. 5c. The SSH drop associated with the NC varies between 2 cm and 15 cm, with a clear seasonal tendency. Greater values are generally observed in winter and smaller

values in summer. The NC core position varies between 10 and 30 km from the coast (30 km in Alberola et al., 1995) with a slight seasonal variation. It is a little closer to the coast in autumn than in winter, in agreement with Niewiadomska et al. (2008) and Sammari et al. (1995), even if these previous studies were not in the Toulon area. The NC width spreads over 10 to 25 km, depending on the season (it is the widest in January and July and the narrowest

in March and April). Previous studies show a NC narrower and faster in winter, it may depend on the NC orientation in relation to the section: a NC not purely perpendicular may artificially increase the current width. In the different diagnostics, the high frequency variability is also important, with some strong peaks. This may be due to intense wind events which induce meanders or eddies in the HF radars area (Guihou et al., 2013). Note that in

August 2013, the NC core shifted until 50 km from the coast, associated with a large width and strong SSH drops (Fig. 5c). It is also visible on Fig. 3 for both the simulation and the radars. We investigated what happened for the corresponding dates, from 25 to 28 August 2013, in both simulated and observed surface currents (not shown). We observed that the NC is then totally deviated to the south and is cut in two parts, with a recirculation loop that

comes from the south-west and blocks the NC flow.

In terms of SSH drop, the NC signature is then generally above the global rms mean error level for the altimetry missions considered here (from Vergara et al., 2019: 2.23/1.66/1.12 cm for Jason-2/SARAL/S3A, respectively). But its width is generally below the scales resolved (from Raynal et al. (2017), Jason satellites can capture offshore dynamical signals down to ~70

km wavelength, SARAL/AltiKa and Sentinel-3 down to 35-50 km). We also know that the observation of near-shore SSH estimates is a technical challenge for altimetry (Vignudelli et al., 2011). In the next section, using the model as the reference, we analyze which part of the NC SSH and current signals are really sampled by altimetry data.


# 4. Observability of the NC in altimetry data: from Jason-2 to Sentinel-3

In this section, we first analyze the general characteristics of the SSH and surface (cross-section) velocity profiles observed along the selected tracks (see section 2.1 and Fig. 2) for

both altimetry data and the model reference (section 4.1). The altimetry data noise issue is then investigated in section 4.2.

## 4.1 SSH and current statistics

We compute the temporal mean and standard deviation of the individual SSH and




corresponding cross-track velocity profiles (using the geostrophic equation) observed along
Jason 2 track 222, SARAL track 302 and Sentinel-3 track 472 (Fig. 6). The corresponding
model estimates at the dates closest to altimetry are also calculated and shown in the same
figures. The model fields are interpolated at the 1-Hz altimetry points along each track (i.e.
every 6-7 km depending on the altimetry mission). Note that here, no spatial filtering is
applied on altimetry data, neither on the SSH nor before computing the geostrophic
velocities, because we want to analyze the resolution capability of raw sea level data. For
Jason-2 and SARAL, the time periods considered are the common periods between
observations and the model simulation: 27/05/2011-01/10/2016 for Jason-2 and 24/03/2013
to 13/03/2016 for SARAL. For Sentinel-3, the common period of model and data availability
is very short (18/06/2016 to 15/03/2017). We then chose to use the total period of Sentinel-3
data availability (18/06/2016 to 14/03/2019) and the period 21/06/2014 to 15/03/2017 for the
model (same length but different years). To estimate the impact of this choice on the results,
we performed a sensitivity analysis by computing the mean current and the mean SSH of the
model (same diagnostics than those on Fig. 6) over different 3-year time periods: over
10/06/2011 - 31/03/2014, 22/06/2012 - 17/03/2015, 08/06/2013 - 29/03/2016. The results are
very similar (not shown), which indicates that in this area the interannual variability does not
have a strong imprint on our results.

The three diagnostics defined in section 3 are considered for each mission - the SSH drop
associated with the NC, the NC width and the distance to the coast of the NC core - and
extended up to 120 km of the coast. The statistics are computed with 195, 32, 36 samples for
Jason-2, SARAL and Sentinel-3, respectively.

We first focus on Jason-2 results. In Fig. 6a, we observe that on average, the raw altimetry
SSH profile agrees fairly well with the model above 20 km from the coast; below this
distance, the two curves diverge with a steeper slope for Jason 2. In this area, the SSH
increase corresponding to the external edge of the NC starts at 60 km to the coast, i.e. further
from the coast than for the 6.14°E transect (located to the west). The 1-Hz altimetry SSH data
stops at 8 km from the coast. SSH standard deviations from altimetry are slightly greater
(between 0.8 and 1.6 cm) than from the model, except at the nearest point to the coast where
the difference reaches 2.2 cm. Figure 6b shows the corresponding mean cross-track velocity
profiles. Jason-2 solution is noisier than the model one. Here again, above 20 km from the
coast the two mean curves agree well but when approaching the coast, the steeper slope
observed in Jason-2 SSH results in too high near-coastal velocity values and then a larger
NC, in comparison to the model. The standard deviation of Jason 2 velocities is about three
times higher than for the model ($0.34$ m s$^{-1}$ against $0.092$ m s$^{-1}$). We also observe that the
current variability tends to decrease near the coast in the model, whereas it increases in the
observations, likely due to nearshore increased altimetry noise.

Figure 6c,d shows the same analysis for SARAL. It should be kept in mind that the 35-day
cycle of SARAL and its shorter lifetime lead to a significantly smaller number of samples to
compute the statistics compared to Jason-2. Figure 6c shows the SSH profiles. Here, 1-Hz


altimetry data stops at 16 km from the coast. The SARAL and model curves have more or less similar slopes but SARAL SSH begins to increase much further from the coast than the simulated SSH (70 km vs 50 km). On the contrary to Jason 2, the SARAL SSH variability is quite similar (std difference of 0.5 cm) to the simulated one near the coast. The corresponding mean velocity profiles have similar shapes, but slightly more spreaded offshore for altimetry (Fig. 6d). The SARAL-derived currents are less noisy than Jason 2 ones but with still greater variability than the model reference (std of 0.16 m s$^{-1}$ for SARAL raw data, and 0.068 m s$^{-1}$ for the model).

Finally, we repeated the process for Sentinel-3 (Fig. 6e,f). As explained before, the model is shifted in time in order to have enough data to compute statistics. In terms of SSH profile (Fig. 6e), Sentinel-3 appears very similar to SARAL (Fig. 6c). SSH increases further south for the observations than for the model, leading to a slightly more offshore extended current. Compared to Jason 2 and SARAL, Sentinel-3 1-Hz data get much closer to the coast (around 1 km), and are also less noisy with SSH standard deviation quite identical to the model near the coast and slightly higher far from the coast. Figure 6f shows that, thanks to its better coastal data coverage, Sentinel-3 captures the NC almost entirely. The current variability remains quite important along the track compared to the model (0.19 m s$^{-1}$ for altimetry against 0.065 m s$^{-1}$ for the model in average) and a huge standard deviation value characterizes the first point near the coast.

From the results of Fig. 6, we computed the time-averaged NC characteristics (SSH drop, NC width and distance to the coast of the NC core). The results are summarized in Table 2. For Jason 2, the NC signature in SSH is significantly stronger than that seen by the model sampled as altimetry: 10.2 cm and 7.2 cm respectively. This is mainly due to the divergence between the model and altimetry SSH near the coast. SARAL is very close to the model: 7.1 cm against 6.8 cm. Sentinel-3 is in between, with a drop of 8.2 cm vs 6.8 cm for the model. The NC width is slightly larger in altimetry than in the model (+6/+5/+1 km for Jason-2/SARAL/Sentinel-3, respectively). In Jason-2 and Sentinel-3, the NC core is located at the same distance to the coast as in the model, but it is located 8 km further from the coast in SARAL. Note also that the NC is better (almost entirely) resolved in Sentinel-3, compared to Jason-2 and SARAL.

## 4.2 The altimetry data filtering issue

In practice, users systematically apply a spatial filter to altimetry SSH data before geostrophic current derivation, an operation that strongly amplifies the measurement noise, as observed in section 4.1. The SSH filtering step is then a key element of altimetry current computation and it is even more true in coastal areas. Consequently, the capability of altimetry to capture mesoscale currents depends on the choice of the filter.

Figure 7 illustrates this noise issue by presenting the time space diagram of SSH derived from



the model and from 1-Hz altimetry raw data along the Jason 2 track 222 in the 120 km close
to the coast. Note that with Jason-2, for the reason explained before, near-shore data are often
missing. If the evolution of both SSH fields is globally similar, we clearly observe noise in
altimetry data as well as larger differences near the coast (i.e. in the first 30 km).

To estimate the best SSH filtering for the derivation of current estimates, we compute the
distribution of the resulting geostrophic velocity values, using raw and low-pass filtered SSH
altimetry data in the 60 km close to the coast. We compare the results to the distribution of
the corresponding model velocities, used here again as a reference. To obtain the filtered
SSH, we tested different cutoff frequencies, ranging from 30 km to 50 km for SARAL and
Sentinel-3 and extending to 70 km for Jason 2. Indeed, Morrow et al. (2017) and Raynal et al.
(2017) showed a greater noise level in Jason 2 which required larger cutoff frequency values.
The histograms of current values are represented in Fig. 8 for Jason 2 track 222, in Fig. 9 for
SARAL track 302 and in Fig. 10 for Sentinel-3 track 472  (altimetry in blue superposed on
the model in pink). Note that for each mission,  the model current values are sampled at
altimetry temporal resolution (10, 35 and 27 days for Jason 2, SARAL and Sentinel-3
respectively) and at the model resolution to investigate the impact of undersampling data
(bottom figures). Table 3 summarizes the statistics derived from the histograms: the median,
the standard deviation, as well as the number of points outside typical current values in this
area and considered as outliers (greater than 0.25 m s$^{-1}$ and smaller than -0.6 m s$^{-1}$. These
values are considered the typical NC velocities).

We first focus on Jason 2. The model reference shows a distribution which tends to be
gaussian. It is centered around -0.15 m s$^{-1}$, with a majority of negative values and is slightly
asymmetric. Jason-2 raw velocity values are almost randomly distributed. When Jason-2 SSH
data are filtered, and as the cutoff wavelength increases, the histogram's distributions change
and get closer to the model ones. Regarding the statistics (Table 3), the too high standard
deviation and too negative median values in the raw Jason-2 data get closer to the reference
with the increase in cutoff wavelength. With a 60 km - filtering, we have the same standard
deviation values in both Jason-2 and model velocities but the median value remains always
significantly lower in Jason-2. The number of outliers is also too large in raw Jason-2 data,
but decreases rapidly with the filtering; it is the closest to the model reference for a 60 km -
filtering. From these results we conclude that Jason 2 currents tend to best converge towards
the model reference with a filtering at 60 km. Beyond this cutoff wavelength, the smoothing
erases the left and right-hand sides of the distribution (Fig. 8) and reduces the variability.
We repeat the same analysis with SARAL (Fig. 9 and Table 3). Note that there are fewer
satellite cycles for SARAL than for Jason 2, so less current data are available to compute
statistics. As a result, the distributions obtained are more complex than for Jason-2. It is
clearly observed when comparing Fig. 9e and f (distributions computed at the model
resolution and at a 35-day resolution). The model histogram is initially centered on -0.07 m s$^{-1}$
with an asymmetric shape and a slight secondary peak around -0.25 m s$^{-1}$. When using the
SARAL temporal resolution, the distribution is more random with a peak around -0.07 m s$^{-1}$.


The raw altimetry solution is less randomly distributed than for Jason 2, also confirmed by a
standard deviation value 2 times smaller than for Jason-2, 0.18 m s$^{-1}$ vs 0.36 m s$^{-1}$ and already
relatively close to the 0.15 m s$^{-1}$ model reference. SARAL tends to converge towards the
model with a filtering of 30 km.

For Sentinel-3, the distribution of the raw altimetry solution has a bimodal shape (Fig. 10a) as
in the model. Its standard deviation is also largely closer to the model reference, compared to
Jason-2 (but slightly less than SARAL, Table 3). The statistics of the altimetry velocities tend
to converge towards the model reference with a 40-50-km cutoff wavelength. One of the
reasons for the slightly bimodal distribution in SARAL and Sentinel-3 may be the track
orientation, quite different from the Jason 2 track which is perpendicular to the NC (Fig. 2e).
Testing different track angles reveals indeed a small second peak (not shown).

# 5. Summary and conclusion

In this study, we have presented a novel method to quantify the SSH signature of a narrow
slope current, the Northern Current in the North Western Mediterranean Sea, and to define its
observability in altimetry data. It is based on a high resolution numerical model, intensively
validated against in situ and HF radars data, and then considered as a reference for satellite
altimetry data analysis. We consider the SSH and related surface geostrophic currents in
parallel, using three nadir-looking radar altimeters that employ different technologies: Jason-
2, SARAL and Sentinel-3. We investigate how the advances in performance of these
altimeters (Vergara et al., 2019) improve the observation of the NC.

We show that in the HF radars covered region the NC has a clear signature in SSH,
characterized by a sea level drop from offshore to the coast, generally centered at ~15-20 km
to the coast with a mean value at 6.14°E of 6.9 ± 2.2 cm and spreading over 18 ± 4 km. In
winter, the SSH drops are generally stronger than in summer and then theoretically easier to
detect for altimeters but in parallel, the NC width also tends to diminish, inducing opposite
effects in terms of observability capability. Sammari et al. (1995) have noticed a variability of
the alongslope component of the NC between 10 and 20 days that altimetry does not allow to
resolve. These results confirm that as a narrow, variable and close to the coast current, the NC
monitoring is an issue for satellite altimetry and that the larger the temporal resolution of the
mission, the greater the difficulty. This is particularly true for SARAL and Sentinel-3,
regardless of the instrument performance.

We then analyze the NC signature in altimetry data in comparison to the model reference.
Jason 2 and SARAL 1-Hz data stop at more than 10 km to the coast, sometimes preventing
observation of the whole NC. Probably thanks to the SAR mode, it is resolved in Sentinel-3,
with data at 1 km to the coast. In average, the SSH drops associated with the NC are always



too high in altimetry, with mean values of 3.6 cm, 0.3 and 1.4 cm larger for Jason 2, SARAL and Sentinel-3, respectively. The mean NC core location is correctly located in Jason 2 and
Sentinel-3 but it is slightly shifted in SARAL (an 8 km difference between the model and observations). In terms of current variability, Sentinel-3 and SARAL have much lower values than Jason-2 (which remain 3.7 times larger than the model reference). These too high values are of course largely explained by the measurement noise, significantly decreasing in the most recent altimetry missions. In section 4.1, we observe how this noise reduction has a
strong impact on the quality of velocity derived from SSH, with the best performance obtained with SARAL data. However, the signal-over-noise ratio remains too large and all satellite SSH data must clearly be filtered before computing currents. By comparing the distributions of altimetry velocity fields derived with different filtering strategies with the model reference, we find that the optimal cutoff wavelength is 60 km, 30 km and 40-50 km
for Jason-2, SARAL and Sentinel-3 SSH data, respectively. Note that these values are slightly lower than the numbers given in Raynal et al. (2017): ~70 km for Jason-2 and 35-50 km for SARAL/AltiKa and Sentinel-3, even if these studies focused on open ocean data. Morrow et al. (2017) also found values similar to Raynal et al. (2017) for Jason 2 and SARAL missions through spectral analysis.

In summary, in terms of coastal circulation studies, the main advantages of Jason missions are the long time series (~30 years if we combine T/P and Jason-1,2,3 data) and its 10-day temporal resolution whereas its measurement noise, its large intertrack distance and its loss of coastal data are an issue. SARAL enables to strongly reduce the data noise and to have a much better spatial coverage but its temporal resolution and its loss of coastal data are negative points. Finally
Sentinel-3 performs the best when approaching the coast and its SAR technique also allows to reduce the noise. However, it has a too low temporal resolution. To ideally address the coastal observability question, future altimetry missions should combine instrumental improvements (Ka band and SAR altimetry as in SARAL and Sentinel-3) and the temporal resolution of Jason or better. But a single altimeter mission can clearly not observe the complex range of
coastal ocean variability and we should certainly define an approach optimized for the coastal ocean in order to mix data from the 9 missions flying simultaneously in 2021.

The method presented here can be easily transposed to other altimetry missions and other dynamical processes than the NC. As an example, we could also focus on eddy observability, studying the size, amplitude and spatial configuration of their signature in SSH, in
comparison to the model reference. More generally, this study illustrates the benefits of using a carefully calibrated high-resolution model as a reference for coastal altimetry studies. It allows overcoming the sparsity of independent observations needed to validate near-shore altimetry observational data. Models can be used as a reference to compare the performance of different altimetry missions, but also to compare or calibrate coastal data processing
strategies. And finally, they provide 3D information on the whole range of ocean parameters that can be related to the sea level variations captured by altimetry.





## Acknowledgements

This study was done with the financial support of the Region Occitanie and the CNES
       through their PhD funding programs. Altimetry data used in this study were developed,
       validated and distributed by the CTOH/LEGOS, France. Glider data were collected and made
       freely available by the Coriolis project (http://www.coriolis.eu.org) and programs that
       contribute. Support was provided by the French Chantier Méditerranée MISTRALS program
(HyMeX and MERMeX components) and by the EU projects FP7 GROOM (grant agreement
       284321), FP7 PERSEUS (grant agreement 287600), FP7 JERICO (grant agreement 262584)
       and the COST Action ES0904 EGO (Everyone's Gliding Observatories). The long-term
       monitoring of the Northern Current is part of the Mediterranean Ocean Observation Service
       for the Environment (MOOSE) with HF radars activities also supported by the EU H2020
infrastructure project JERICO-NEXT (2015-2019) and actually by EU Interreg Marittimo
       program SICOMAR- PLUS. We thank the Parc National de Port-Cros (PNPC), "Association
       Syndicale des Propriétaires du Cap Bénat" (ASPCB) and the Group Military Conservation
       and the Marine Nationale for hosting our radar installations. The simulation was performed
       using the HPC  CALMIP platform  under grant P09115 and GENCI and CINES (Grand
Equipement National de Calcul Intensif, project A0040110088). The SYMPHONIE model is
       distributed by the SIROCCO group (https://sirocco.obs-mip.fr).

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



**Table 1: Characteristics of the altimetry datasets used in this study as a function of the satellite mission.**

|  | Altimetry mission | | |
|---|---|---|---|
|  | **Jason 2** | **SARAL** | **Sentinel-3** |
| **Track used** | 222 | 302 | 472 |
| **Data period** | June 2008 - October 2016 | April 2013 - May 2016 | June 2016 - May 2019 |
| **Intertrack distance in the NW MedSea** | 230 km | 58 km | 78 km |
| **Temporal resolution** | 10 days | 35 days | 27 days |
| **Radar technology** | Conventional LRM altimetry - Ku band | Conventional LRM altimetry - Ka band | SAR altimetry - Ku band |
| **Along-track resolution at 1-Hz** | 5.8 km | 7.5 km | 6.7 km |






**Table 2: SSH drop, NC width and distance to the coast computed for the Jason 2 track 222, the SARAL track 302, the Sentinel-3 track 472 and SYMPHONIE sampled as 1-Hz altimetry.**

| Dataset | SSH drop (cm) | NC width (km) | Distance to the coast of the NC core (km) |
|---|---|---|---|
| Jason 2 track 222 | 10.2 | 33 | 27 |
| SYMPHONIE | 6.6 | 27 | 27 |
| SARAL track 302 | 7.1 | 25 | 20 |
| SYMPHONIE | 6.8 | 20 | 12 |
| Sentinel-3 track 472 | 8.2 | 29 | 17 |
| SYMPHONIE | 6.8 | 28 | 17 |





45

**Table 3: Statistics corresponding to the distributions shown on Fig. 8, 9 and 10.**

| | Mission | standard deviation | median | number of points >0.25m s$^{-1}$ or <-0.6 m s$^{-1}$ |
|---|---|---|---|---|
| **model (daily)** | Jason 2 | 0.14 (0.14) m s$^{-1}$ | -0.17 (-0.16) m s$^{-1}$ | 6 (16) |
| | SARAL | 0.15 (0.14) m s$^{-1}$ | -0.16 (-0.16) m s$^{-1}$ | 0 (16) |
| | Sentinel-3 | 0.13 (0.13) m s$^{-1}$ | -0.17 (-0.16) m s$^{-1}$ | 0 (1) |
| **raw** | Jason 2 | 0.36 m s$^{-1}$ | -0.20 m s$^{-1}$ | 342 |
| | SARAL | 0.18 m s$^{-1}$ | -0.22 m s$^{-1}$ | 7 |
| | Sentinel-3 | 0.23 m s$^{-1}$ | -0.19 m s$^{-1}$ | 18 |
| **30 km** | Jason 2 | 0.23 m s$^{-1}$ | -0.21 m s$^{-1}$ | 104 |
| | SARAL | 0.14 m s$^{-1}$ | -0.19 m s$^{-1}$ | 1 |
| | Sentinel-3 | 0.17 m s$^{-1}$ | -0.20 m s$^{-1}$ | 8 |
| **40 km** | Jason 2 | 0.19 m s$^{-1}$ | -0.21 m s$^{-1}$ | 52 |
| | SARAL | 0.13 m s$^{-1}$ | -0.19 m s$^{-1}$ | 1 |
| | Sentinel-3 | 0.14 m s$^{-1}$ | -0.20 m s$^{-1}$ | 4 |
| **50 km** | Jason 2 | 0.16 m s$^{-1}$ | -0.21 m s$^{-1}$ | 15 |
| | SARAL | 0.11 m s$^{-1}$ | -0.19 m s$^{-1}$ | 0 |
| | Sentinel-3 | 0.13 m s$^{-1}$ | -0.20 m s$^{-1}$ | 3 |
| **60 km** | Jason 2 | 0.14 m s$^{-1}$ | -0.20 m s$^{-1}$ | 9 |
| **70 km** | Jason 2 | 0.12 m s$^{-1}$ | -0.20 m s$^{-1}$ | 1 |




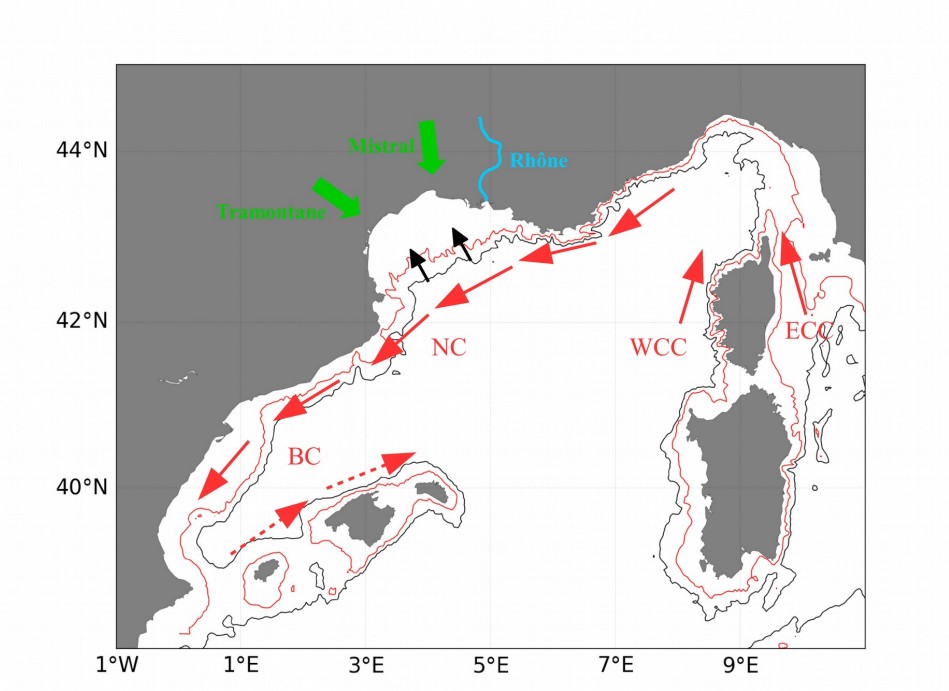

**Figure 1: Schematic circulation in the Northwestern Mediterranean Sea. Red arrows indicate the main currents; black arrows indicate the intrusion in the Gulf of Lion. 200 m (red line) and 1000 m (black line) isobaths are also shown.**

775







**Figure 2: Maps illustrating the location of the observations used in this study as well as the spatial model coverage. (a) Amplitude and vectors of mean surface currents from the HF radars**
780    **near Toulon; the red line shows the transect used in the study. Altimetry tracks in the Western Mediterranean Sea for (b) Jason 2 ; (c) SARAL ; (d) Sentinel-3. For each mission, the tracks used in the study (track 222 for Jason 2 ; track 302 for SARAL ; track 472 for Sentinel-3) are indicated in bold. The HF radars region and the gliders transect are represented in blue. (e) Mean surface currents from the SYMPHONIE model for the period 18/05/2011-31/03/2017. The**
785    **satellite tracks are represented in black.**






55

**Figure 3: Mean zonal current velocities (a) along a meridional section located at 6.14° E for the simulation in blue and the HF radars in green ; (b) along the Nice-Calvi line for the simulation in blue and the gliders in green. The blue envelope represents the standard deviation at each point. The distance is referenced to the coast. Time space diagrams of (c) the zonal current component along a meridional section located at 6.14° E given by the HF radars (top panel) and the simulation (middle panel) ; (d) the geostrophic current for the gliders (top panel) and the simulation at the glider temporal resolution (middle panel). Lower panels of (c) and (d) show the differences between the observations and the simulation.**






**Figure 4 : Maps of the mean current value perpendicular to the tracks derived (a) over the whole region and (c) for a zoomed-in view. (b) and (d) same as (a) and (c) for the standard deviation of currents. The top figure of each panel represents currents from glider, HF radars and altimetry data and the bottom figure from the SYMPHONIE model interpolated at the dates and points of every instrument over the period March 2013-October 2014. 200 m (red line) and 1000 m (black line) isobaths are also shown. Top figures are taken from Carret et al. (2019).**




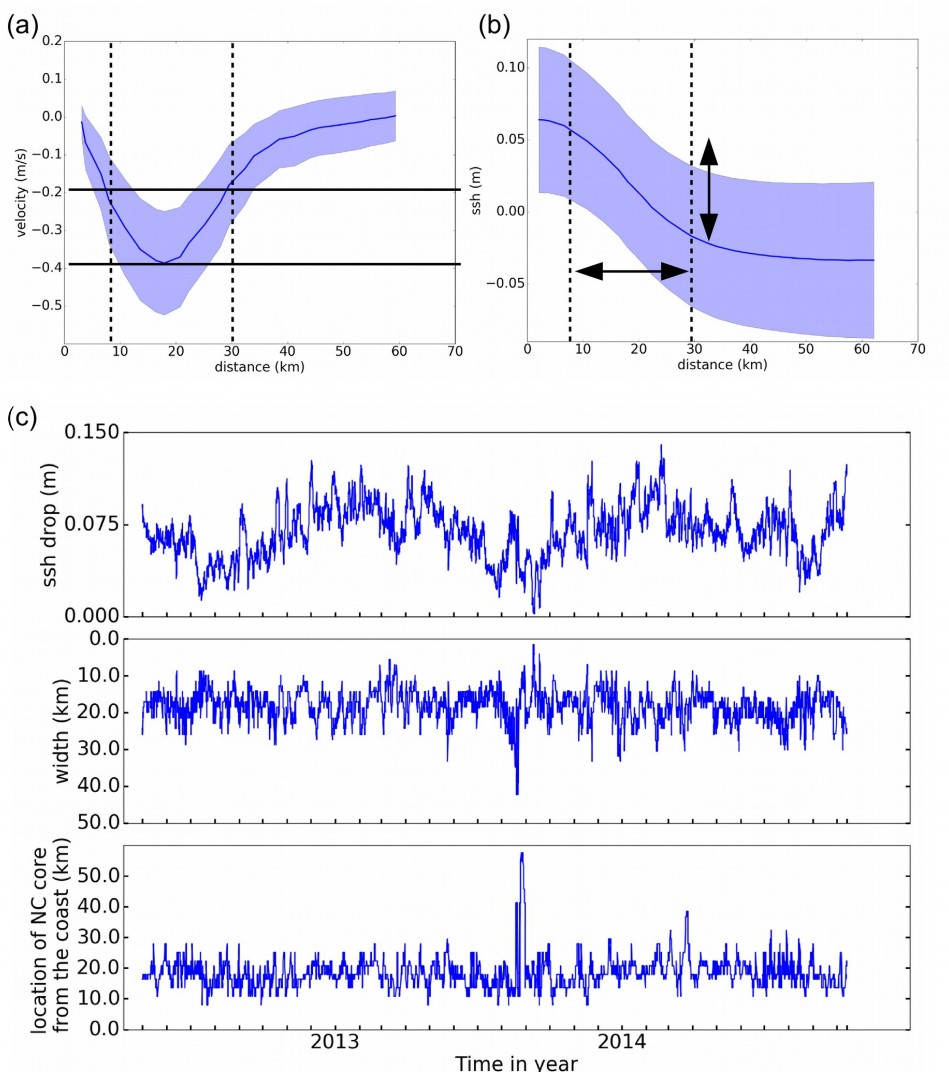

**Figure 5:** Time averaged (a) current velocities and (b) SSH along a meridional section located at
6.14° E for the SYMPHONIE model. (c) Time series of the SSH drop (in m, upper panel), width
(in km, middle panel) of the NC , and location of the NC core as a function of the distance to the
coast (in km, lower panel). The blue envelope in (a) and (b) represent the standard deviation at
each point.


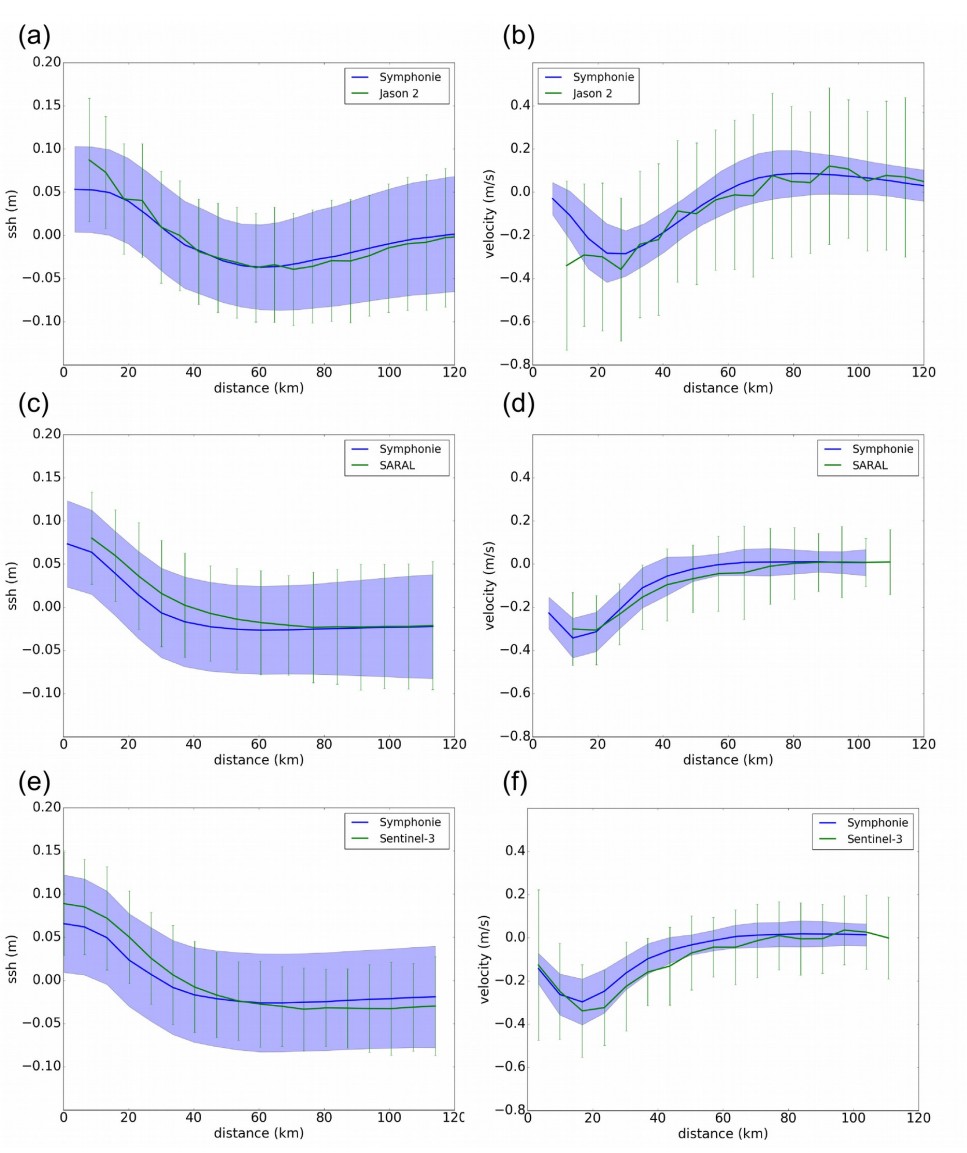

**Figure 6: Mean (a), (c), (e) SSH and (b), (d), (f) current velocities along (a), (b) Jason 2 track 222; (c), (d) SARAL track 302; (e), (f) Sentinel-3 track 472 for the model in blue and altimetry raw data in green. The blue envelope and green bars represent the standard deviation at each point. The distance is referenced to the coast.**

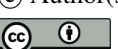


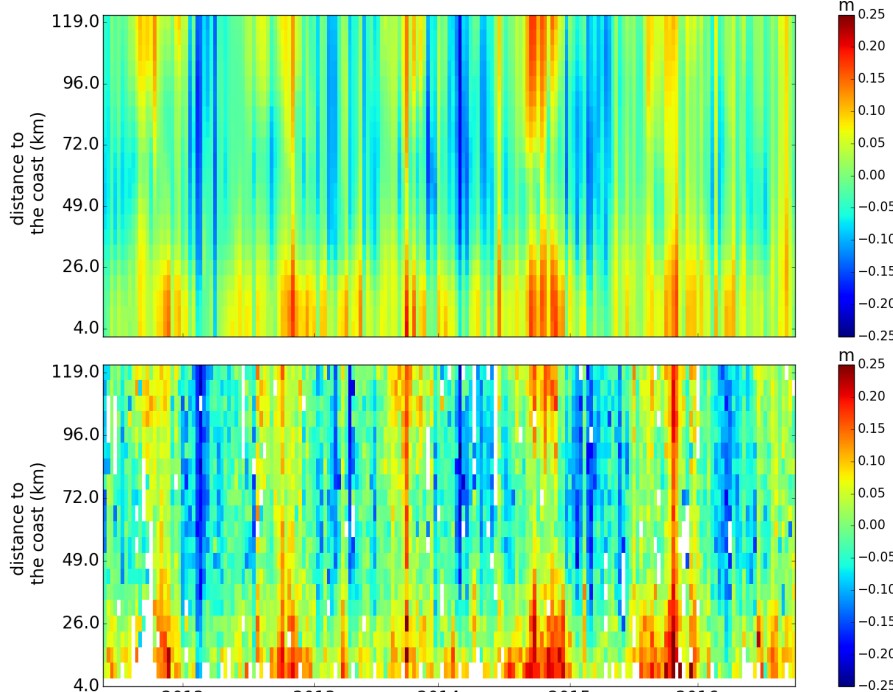

**Figure 7: Time space diagrams of SSH along the Jason 2 track 222 for the model (upper panel) and for Jason 2 (lower panel), as a function of the distance to the coast.**






65





**Figure 8: Distribution of the current values along the Jason 2 track 222 and over the first 60 km to the coast for (a) raw altimetry data and (b),(c),(d),(e),(f) low-pass filtered altimetry data with different cutoff frequencies indicated in the panels. Altimetry distributions (in blue) are**

820 **superimposed on the corresponding model distribution (in pink). The latter is computed for the Jason 2 temporal resolution (g) and for the model resolution (h)**





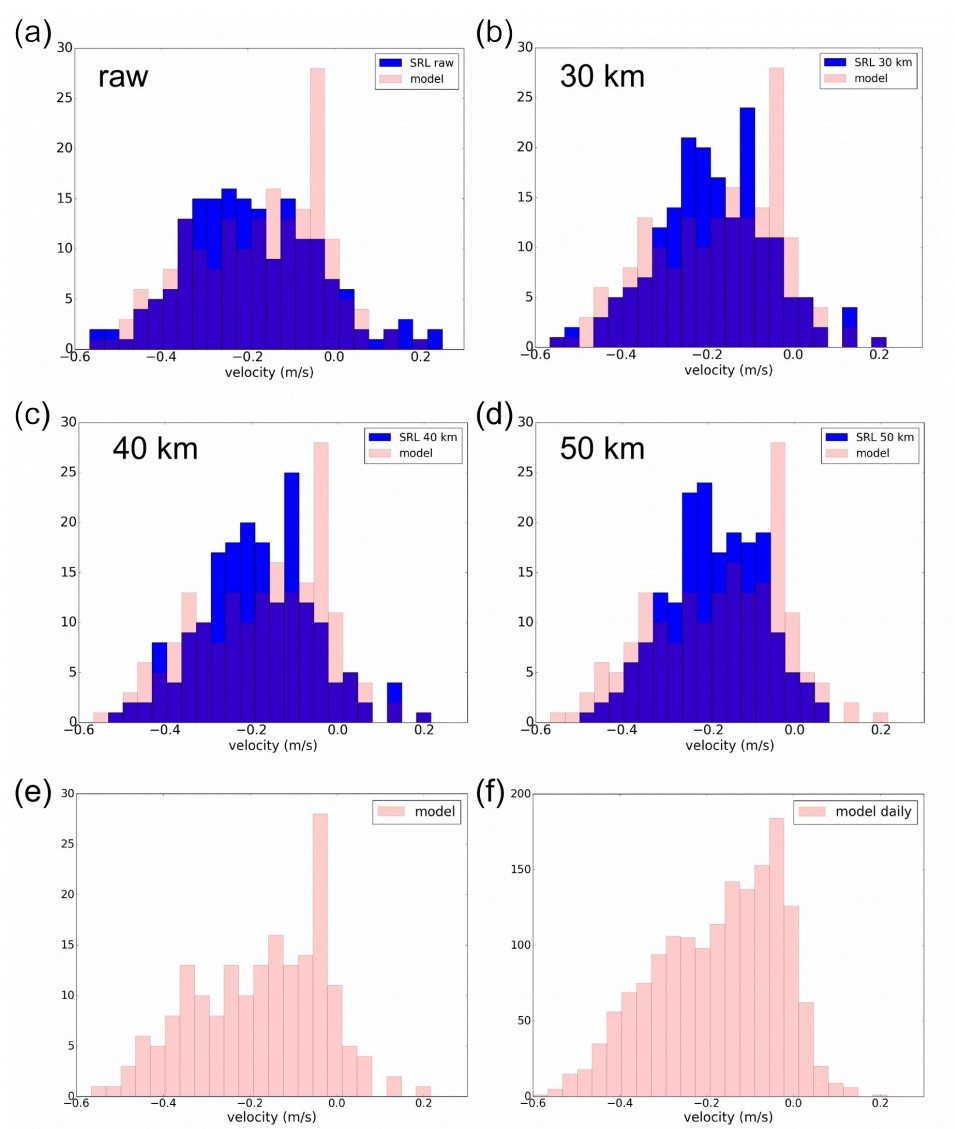

**Figure 9: Distribution of the current values along the SARAL track 302 and over the first 60 km to the coast for (a) raw altimetry data and (b),(c),(d),(e),(f) different filters indicated on each panel. Altimetry distribution (in blue) is superimposed on the corresponding model distribution (in pink). The latter is computed for the SARAL temporal resolution (g) and for the model resolution (h)**

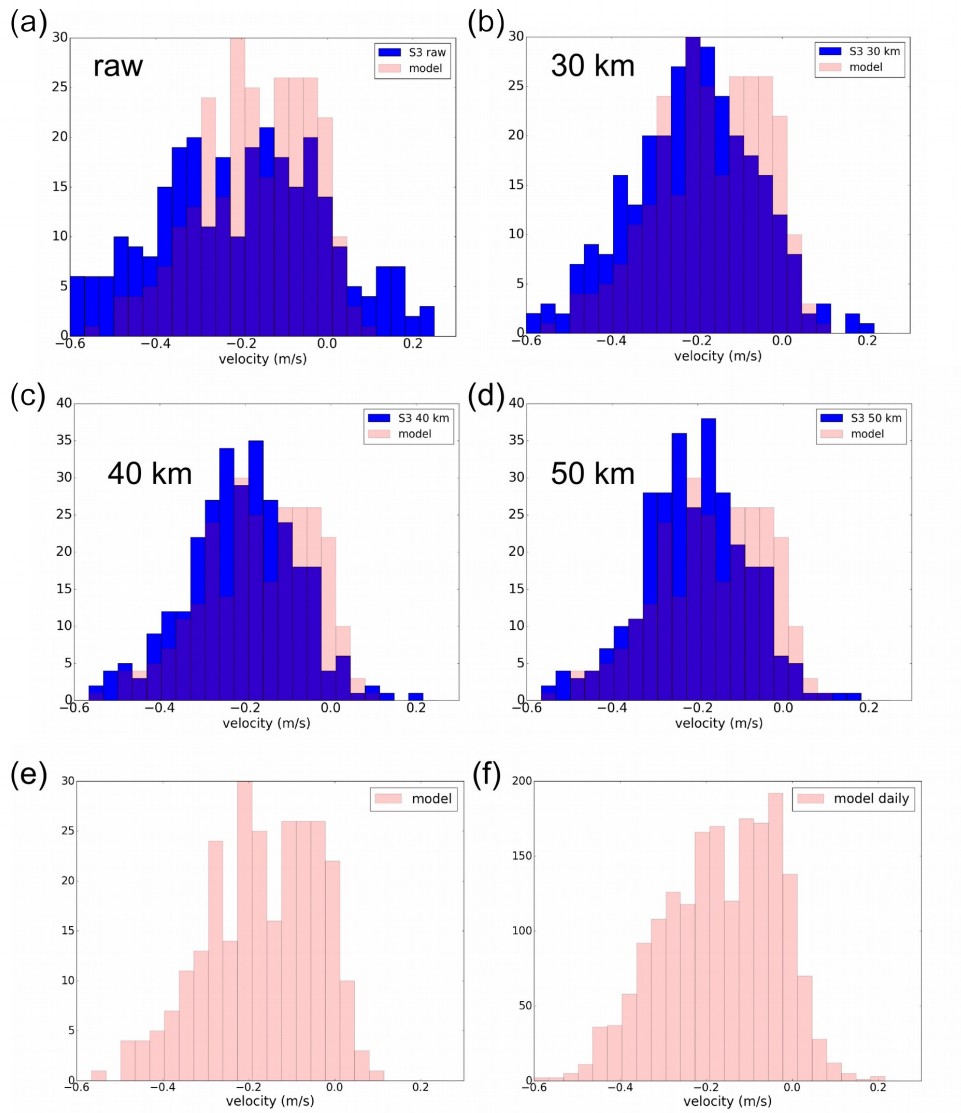

**Figure 10: Distribution of the current values along the Sentinel-3 track 472 and over the first 60 km to the coast for (a) raw altimetry data and (b),(c),(d),(e),(f) different filters (in blue). Altimetry distribution (in blue) is superimposed on the corresponding model distribution (in pink). The latter is computed for the Sentinel-3 temporal resolution (g) and for the model resolution (h)**