# Peer review of "Assessing the capability of three different altimetry satellite missions to observe the Northern Current by using a high-resolution model"

_Ocean Science, 2022_

## Author Comment (AC1)

**Reply to reviewer #1**

**General Comments**

The manuscript evaluates the ability of three altimeter missions (i.e. JASON2, SARAL/AltiKa, Sentinel-3A) data to capture the Northern Current sea level signature (in terms of SSH drop, NC width and distance to the coast) in the coastal ocean, using the high-regional model SYMPHONIE as a reference. Authors have previously assessed the model against High-Frequency (HF) radar and glider data. Findings show the importance of applying spatial filters to altimeter data before computing the geostrophic currents in order to obtain a better agreement with the model. Authors also conclude that the combined effects of instrumental improvements (that reduce the noise and the loss of coastal data), the long-term time series and the higher temporal resolution are essential to enhance coastal features observability.

The manuscript could be better structured in general terms to faithfully reflect quality research, making it more readable, cohesive and concise. On the one hand, I did regret the lack of detail in the section describing the methodology and the lack of a specific section with results and discussion. On the other hand, I felt that there was some unnecessary repetition of the three NC signature diagnostics (e.g. mean NC core location; width and distance to the coast) and of the recent progress made on both altimetry technologies and processing techniques.

Since the ability of each altimeter mission to observe the coastal features (e.g. Northern Current) was expected to be also investigated, I would recommend to include a detailed discussion on how the latest advances in sensors (e.g. in terms of spatial resolution, data accuracy, ionosphere effects, data noise, etc.) and data processing have contributed to improve it.

In addition, considering that the High Frequency Radars (HFR) are emerging as a valuable asset of coastal observing systems, being able to monitor surface currents at unprecedented spatio-temporal scales over wide coastal areas, I think it will be worth it to highlight the benefits of using high-resolution models (instead of HFRs measurements) as a reference (i.e. ground truth) to assess the altimeter data in coastal areas.

It is obvious that substantial effort has been put in this research and I am convinced that this study could help the coastal altimetry community to gain a better understanding of the current performance, limitations and further steps to extend the capabilities of current altimeters closer to the coastal zone.

Therefore, I would strongly encourage authors to resubmit their manuscript after carefully considering all comments below:

We thank the reviewer for these comments that have been taken into account in order to improve the manuscript. Please find below some detailed answers to all your points in red. The manuscript has been restructured, some sections were developed and others deleted.

**Specific comments**

- Title: Authors are invited to consider revising it to clearly reflect the content, with as many significant terms as possible: i) including the use of the high-resolution model as a reference for assessing the altimetry data, in order to highlight the novelty of the methodological study and; ii) specifically mentioning the particular coastal circulation feature analyzed in the study, the so-called Northern Current.
    - Suggestion 1: "Assessment of the Northern Current sea level signature observed by three altimetry satellite missions using a high-resolution model as a reference"
    - Suggestion 2: "Comparison of JASON-2, SARAL/AltiKa and Sentinel-3 data with a high-resolution model to assess their capability to observe the Northern Current"
    - Suggestion 3: "Assessing the capability three different altimetry satellite missions to observe the Northern Current by using a high-resolution model"

Reply: Thanks for these suggestions, we agree and changed the title to suggestion 3 "Assessing the capability of three different altimetry satellite missions to observe the Northern Current by using a high-resolution model"

- Abstract: could describe in a more concise way the motive and objective of the research, the methodology used, the main findings and the conclusions.
    - The main objective of the study should be clarified (particularly from L18-L19).
      Reply: We agree, the abstract has been completely rephrased to be more compact and clearer (see below).
    - Avoid the use of acronyms (e.g, HF, LRM, SAR), unless they are previously defined.
      Reply: It has been done.
    - Avoid unnecessary details of the methodology in the abstract (L26).
      Reply: We removed the sentence "The data from all missions were processed with the coastal-specific X-TRACK strategy."
    - Please, double check all values (L28-L42) since some inconsistencies have been found along the manuscript (e.g. 50-, 30- and 40-km cutoff wavelengths would probably be 60-, 30- and 40/50-km, as concluded in section 4; 18 ± 4 km would probably be 18 ± 9 km, etc.).

      Reply: It has been checked and corrected.

The abstract is now:

"Over the last three decades, satellite altimetry has observed Sea Surface Height variations, providing a regular monitoring of the surface ocean circulation. Altimetry measurements have an intrinsic signal-to-noise ratio that limits the space scales of the currents that can be captured. However, the recent progress made on both altimetry sensors and data processing allow us to observe smaller geophysical signals, offering new perspectives in coastal areas where these structures are important.

In this methodological study we assess the ability of three altimeter missions with three different technologies to capture the Northern Current and its variability (North Western

Mediterranean Sea): Jason2 with its Ku-band Low Resolution Mode altimeter, SARAL/AltiKa with its Ka-band Low Resolution Mode altimeter and Sentinel-3A and its Synthetic Aperture Radar altimeter. Therefore, we use a high-resolution regional model as a reference.

We focus along the French coast of Provence where we first show that the model is very close to the observations of HF radars and gliders in terms of surface current estimates.

In the model, the Northern Current is observed 15-20 km to the coast on average, with a mean core velocity of 0.39 m s$^{-1}$. Its signature in sea level consists of a drop whose mean value at 6.14°E is 6.9 cm extending over 20 km. These variations show a clear seasonal pattern, but high frequency signals are also present most of the time. In comparison, in 1-Hz altimetry data, the mean sea level drop associated with the Northern Current is overestimated by 3.6 cm for Jason 2, but significantly less with SARAL/AltiKa and Sentinel-3A: 0.3 cm and 1.4 cm, respectively. In terms of corresponding sea level variability, Jason 2 and SARAL altimetry estimates are larger than the model reference (+1.3 cm and +1 cm, respectively) whereas Sentinel-3A shows closer values (-0.4 cm). When we derive geostrophic surface currents from the satellite sea level variations, without any data filtering, in comparison to the model, the standard deviation of velocity values are also very different from one mission to the other: 3.7 times too large for Jason-2, but 2.4 and 2.9 times too large for SARAL and Sentinel-3A, respectively. When low-pass filtering altimetry sea level data with different cutoff wavelength, the best agreement between the model and the altimetry distributions of velocity values are obtained with a 60 km, 30 km and 40-50 km cutoff wavelength for Jason-2, SARAL and Sentinel-3A data, respectively. This study shows that using a high resolution model as a reference for altimetry data allows us not only to illustrate how the advances in the performances of altimeters and in the data processing improve the observation of coastal currents but also to quantify the corresponding gain."

- Introduction:
  - Provides an overview of the current efforts that have been made to extend the capability of the altimeters closer to the coastal zone. However, it is recommended the addition of further details about the emerging altimetry technologies, sensors and data processing techniques aiming to address this challenge for satellite altimetry observations.
  - Reply: Done. The end of the first paragraph of section 1 in now: "However, during past years, new altimetry techniques have emerged: such as the use of the Ka-band LRM frequency with the SARAL/AltiKa mission (2013+), the adoption of the Synthetic Aperture Radar (SAR) mode with CRYOSAT-2 (2010+), Sentinel-3A,B (2016+, 2018+) and Sentinel-6 (2020+) and a Ka-band Radar Interferometer (KaRIn) with SWOT (launched in Dec. 2022). In addition, improvements in re-tracking of radar waveforms and a better characterisation and removal of geophysical corrections such as atmospherical effects or tidal signals have all served to improve the precision of the data retrieved. All these progress have led to a

significant gain in observability of the fine scale ocean structures in general and of the coastal features in particular (Birol et al., 2021; Morrow et al., 2017; Verron et al., 2018)."

○ Provides a complete description of the main characteristics and seasonal variability of the Northern Current. Authors are encouraged to highlight the limitations of prior research studies addressing the contribution of along-track satellite altimetry to study the NC variability to help lay a foundation for understanding the research problem investigated by the authors.

Reply: The main characteristics of the seasonal variability of the NC have been completed. Concerning the limitations of prior research studies, we have added a sentence at the end of the second last paragraph, now: "In the past, the NC variability has been intensively studied with in situ observations and models: mesoscale fluctuations at 3-6 days and 10-20 days in Sammari et al. (1995) ; month-long eddies associated in Casella et al. (2011) and Hu et al. (2011) and day-long eddies in Schaeffer et al. (2011). Birol et al. (2010) have highlighted the contribution of along-track satellite altimetry to study the NC seasonal variability. Since then, other altimetry studies have used such data to investigate the NC circulation as well as the recirculation and associated meanders (case studies in Borrione et al., 2019; Morrow et al., 2017; Pascual et al., 2015). But none of them have clearly quantified the observation limit (in both space and time), probably for lack of independent sea level and/or current data sets to do so."

● The datasets and the methodology are partially outlined. Further details about data processing, data availability and data quality procedures are required. Additionally, the methodology used to validate the altimetry data vs. the model is missing in this section, being partially detailed in section 3 (L321-336) and in sections 1 (L383-410) and 4.2 (L466-483) for both spatial unfiltered and filtered data comparison, respectively. Authors should consider to:
   ○ Replace the section 2 title with "Data and methods"
   Reply: We replace the section 2 title only with "Data" as we choose to develop in more details the methods in section 3 rather than adding a specific section.
   ○ Add a completed description of the section 2, including a paragraph explaining that: "The different observing platforms (i.e. HFR, gliders and altimeter missions) and the high-resolution model are described in sections 2.1 and 2.2, respectively, while the model assessment methodology versus HF radar and glider data is detailed in section 2.3". The description of the methodology to validate the altimetry data by using the model as a reference could perhaps be included in this section too.
   Reply: It has been done.
   ○ Clearly specify the post-processing steps applied to be able to compare the HFR velocities with altimeter geostrophic velocities in section 2.1a (e.g. filter out high frequency signals; temporal average; interpolation; etc.). HFR data quality control procedures and data

availability (e.g. URL, DOI) should be included.

Reply: We thank the reviewer for this comment. The DOI has been added and the reader is now referred to the reference in which all the information concerning the HFR data post-treatment is described. We have also added the following sentence: "Note that this data processing removed part of the high-frequency currents, not captured by altimetry that observe only geostrophic currents.".

Note however that in this paper we don't compare HFR velocities with altimeter geostrophic velocities. The HFR velocities are only used for the model validation and then in terms of absolute current: we compare the total surface velocity along a section located at 6.14°E, with the same temporal resolution: every day. To clarify this point we added a sentence in the section 2.3: "The model equivalent is extracted along this section with the same spatial and temporal resolutions as the HF radars. Daily outputs for the model during the HF radars period are used." and we have removed other indications scattered in other sections. We also added a description of how we compared gliders data to the model.

○ Go into detail on how the improvements in altimeters (section 2.1c) results in more accurate measurements, leading to a better characterization of coastal processes. Authors are requested to provide further information in terms of better vertical resolution (due to the enhancement of the bandwidth); improvement of the spatial resolution (thanks to the Ka-band smaller footprint); less affection of the ionosphere (lower for Ka-band); impact of track angles orientation; etc.

Reply: In this section, the first paragraph has been rephrased accordingly. Now: "Jason 2 was launched in June 2008 and was in the same orbit up to October 2016. It is based on the conventional Low Resolution Mode (LRM) altimeter operating in the Ku-band and has a 10-day repetition cycle. SARAL, launched in February 2013, provides a shorter data time series (~3 years) because it moved to a drifting orbit in July 2016. It has a 35-day repeat observation cycle. Its Ka-band LRM altimeter (called AltiKa) has a smaller footprint than the Ku-band instruments: ~4 km radius against 5-7 km. The corresponding lower data noise allows to capture smaller spatial scales than Jason 2 (Verron et al., 2018). The Ka-band is also less affected when crossing the ionosphere and provides a better estimation of the surface roughness. Sentinel-3 was launched in February 2016. With its SAR altimeter, its footprint is even more reduced in the along-track direction, compared to LRM altimeters: ~0.3 km. It has a 27-day repeat observation cycle."

In the second paragraph wa have also added: "As alongtrack altimetry data allows to derive only the across track currents, through the geostrophic assumption, the angle of the tracks with respect to the current vein has a major impact on the current capture: the less

perpendicular the track, the less realistic its amplitude. Concerning SAR altimeters the observation of a current perpendicular to the track will benefit from the corresponding increase in resolution."

- ○ Include and cite prior SYMPHONIE model assessment studies and error estimations of surface currents to further demonstrate its ability to reproduce the main characteristics of the circulation in the study area and its variability (e.g. based on the results obtained by Estournel et al., 2003) to reinforce its role as ground truth (e.g. reference).
  Reply: References have been added: "Validation studies of SYMPHONIE currents over the Gulf of Lion have been carried out by comparison with various instruments on different hydrological structures and meteorological situations: VHF radars on the Rhone plume (Estournel et al., 2001), hull-mounted ADCP (Estournel et al., 2003) in prevailing northerly winds, fixed ADCP (Mikolajczak et al., 2020), and glider drift (Gentil et al., 2022) during easterly storms."
- ○ Please, add an additional Table showing the quantitative assessment based on statistics (e.g. average and standard deviations) of the different NC signature diagnostics (i.e. NC maximum amplitude, NC core location, NC width) provided by HF radars and glider Nice-Calvi transect and compared to the SYMPHONIE model, similar to Table 2.
  Reply: We added this table in the paper.

- Signature of the NC on sea level.
  - ○ Authors are encouraged to provide further information or reference(s) for the selected criteria used to define the width of the NC (i.e. length of the section around the NC core).
    Reply: We provided a reference and added "This criterion offers the advantage of not being impacted by seasonal differences in the NC amplitude."
  - ○ It might be worth considering the description of the extraordinary event in August 2013, which led to the blocking of the NC flow (as mentioned in L359-365), in order to prove the model reliability for describing both, average and extraordinary NC events.
    Reply: We added "The good agreement between the model and the HF radars during this extraordinary event is a proof of the model reliability to reproduce the high frequency variability of the NC."
- Observability of the NC in altimetry data
  - ○ I would suggest a slight rewording of the section 4 for clarity: 'In this section a quantitative assessment of the NC sea level signature (in terms of SSH drop, NC width and distance to the coast) is performed for the three altimeter missions and the reference model. We consider both unfiltered and filtered 1 Hz SLA data for the computation of geostrophic velocities in sections 4.1 and 4.2, respectively, to analyze the importance of applying spatial filters to altimeter data in order to obtain a better agreement with the model'.
    Reply: Thanks, it has been done.
- Summary and Conclusion
  - ○ Results do not sufficient support the interpretations and conclusions

included in this section.

Reply: The summary and conclusion has been largely rephrased (and shortened). Now: "In this study, we have presented a novel method to quantify the SSH signature of a narrow slope current, the NC in the NWMED, and to define its observability in altimetry data. It is based on a high resolution numerical model, intensively validated against in situ glider and HF radars data, and then considered as a reference for satellite altimetry data analysis. We consider the SSH and related surface geostrophic currents in parallel, using three nadir-looking radar altimeters that employ different technologies: Jason-2, SARAL and Sentinel-3.

We show that in the HF radars covered region the NC has a clear signature in SSH, characterized by a sea level drop from offshore to the coast, generally centered at ~15-20 km to the coast, with a mean value at 6.14°E of 6.9 cm and spreading over 18 km. In winter, the SSH drops are generally stronger than in summer and then theoretically easier to detect for altimeters. The NC is also clearly associated with high frequency variability (sections 2.3 and 3). These results confirm that as a narrow, variable and close to the coast current, the NC monitoring is an issue for satellite altimetry. It is also important to note here that, despite the intrinsic performances of the instruments, the temporal resolution of the missions is a very important factor for the observation of coastal currents like the NC. And on this point the advantage is for Jason-2, compared to SARAL and Sentinel-3 missions.

We then analyze the NC signature in altimetry data in comparison to the model reference. Jason 2 and SARAL 1-Hz data stop at 8 and 16 km from the coast, respectively, sometimes preventing observation of the whole NC. Probably thanks to the SAR mode, it is better resolved in Sentinel-3, with data at 1 km to the coast. In average, the SSH drops associated with the NC are always overestimated in altimetry, with mean values of 3.6 cm, 0.3 and 1.4 cm larger for Jason 2, SARAL and Sentinel-3, respectively. The mean NC core location is correctly located in Jason 2 and Sentinel-3 but it is slightly shifted in SARAL (an 8 km difference between the model and observations). In terms of current variability, all altimetry missions show much higher values than the model, because of the measurement noise. But this overestimation decreases significantly from Jason-2 (3.7 times larger) to the more recent Sentinel-3 and SARAL missions . The values closest to the model reference are obtained with SARAL (2.4 times larger, against 2.9 for Sentinel-3). However, the noise remains too large and all satellite SSH data must clearly be filtered before computing currents. By comparing the distributions of altimetry velocity fields derived with different filtering strategies with the model reference, we find that the optimal cutoff wavelength is 60 km, 30 km and 40-50 km for Jason-2, SARAL and Sentinel-3 SSH data, respectively.

In summary, to ideally address the coastal observability question, future altimetry missions should combine instrumental improvements (Ka band and SAR altimetry as in SARAL and Sentinel-3) and the temporal resolution of Jason or better. Another approach would be to better optimize the use of data from the 9 altimetry missions flying

simultaneously in 2023.
The method presented here can be easily transposed to other altimetry missions and other dynamical processes than the NC. As an example, we could also focus on eddy observability, studying the size, amplitude and spatial configuration of their signature in SSH, in comparison to the model reference. Using a carefully calibrated high-resolution model as a reference for coastal altimetry studies allows to overcome the sparsity of independent observations to validate near-shore altimetry data. Models can be used as a reference to compare the performance of different altimetry missions, but also of different coastal data processing strategies. They also provide 3D information on the whole range of ocean parameters that can be related to the sea level variations captured by altimetry."

○ A previous detailed discussion on how the latest advances in sensors (e.g. in terms of spatial resolution, data accuracy, ionosphere effects, data noise, etc.) and data processing have contributed to improve the observability of the coastal features is required to be able to include the sentence between L536-L537 in this section.
Reply: This sentence has been removed.

● References
○ Review the reference list; some of the references included (e.g. Borrione) are missing in the text.
Reply: It has been done.
○ Please, wrap the text between L685-L750.
Reply: It has been done.

● Tables and Figures:
○ As a general comment for the entire section: Please, clearly specify the variable which is being shown in the figures (e.g. geostrophic velocity, surface current velocity, eastward -zonal- or northward -meridional- component of the current) as well as the time period used to calculate the average values.
Reply: We have completed all figure legends to clearly specify the variable and the time period.
○ Table 1: Please, include the sampled used (as mentioned in section 4) and the global SSH RMS for each mission (as included in section 3).
Reply: It has been included.
○ Table 2: add a 4th column including the dates of the analyzed period for each mission (as included in section 4).
Reply: It has been done
○ Table 2 – caption: include "Northern Current SSH signature based on averaged SSH drop, width and distance to the coast computed for…."
Reply: it has been done
○ Table 3: Please, add further details for every datasets in the first column (e.g. raw or unfiltered altimetry data; X-km cutoff wavelength for altimetry data), add the units in the column header instead of in every row.
Reply: it has been done
○ Fig. 1: Please, add an inset map of the NWMed (similar domain as in Fig 2. e) to provide a regional reference and indicate by a black square the extent of the main map (i.e. the one currently shown). Please, include basic geographic features mentioned in the

manuscript (e.g. Toulon, Nice, Calvi, Gulf of Lion, Ligurian Sea, Balearic Islands, Mediterranean Sea, France, Spain, etc.) for helping readers to locate the area discussed in the text.

Reply: We have added an inset map and included basic geographic features.

○ Fig. 1 – caption: "Map of the NW Mediterranean study area, with inset map showing the location of the main map (outlined by a black box). …Both maps contain labels to geographic features mentioned in the text…". Please, describe the acronyms (e.g. NC=Northern Current; BC=Balearic Current, etc.).

Reply: It has been done.

○ Fig. 2: a) Please, zoom in the map to allow the visualization of the current vectors and add the location of the HFR antennas and Toulon; b), c), d) draw altimetry tracks with dashed lines, with the exception of the track used in the study (to be highlighted in bold).

Reply: It has been done.

○ Fig. 2- caption: Please, remove blank spaces before the semi-colons. Consider to: replace "Amplitude and vectors of mean surface currents" with "Mean surface current velocity map"; add the period of the HFR temporal average; replace "HF radars region" with "HF radars coverage area"; add "Nice-Calvi glider transect"; replace "Mean surface currents from the SYMPHONIE" with "Mean surface current intensity from the SYMPHONIE", since the map only shows the current speed.

Reply: it has been done.

○ Fig. 3: a & b) Please, set the same maximum and minimum values in both OY axes. c & d) Please, use the same colorbar limits for all the Hovmöller diagrams. Please, consider to replace the jet colormap in the bottom panels with a blue-white-red colormap such that zero is always color-coded in white to better highlight the differences between the model and the HFR or the glider data. Please, include the number of the month for each year (or the name of the seasons) in the OX axis to better identify the seasonal variability.

Reply: All these comments have been taken into account, and the figure changed accordingly.

○ Fig. 3- caption: Please add the meaning of the green bars (i.e. standard deviation for the satellite data) and replace "Time space diagrams" with "Hovmöller diagrams". Please remove "The distance is referenced to the coast", since any distance is provided in the figure.

Reply: it has been done

○ Fig. 4: Please, remove a & b panels (for the whole region) and keep c & d in the figure. Altimetry data over the HFR footprint area should be highlighted, e.g. increasing the size of the dots. If the authors decide to keep a & b panels, please, use the same colorbar limits in all panels.

Reply: We have removed this figure in the final version.

○ Fig. 5: a) It is not clear which is the difference with Fig. 3a. Is the mean zonal current velocity? Please, clarify it. C-bottom panel) Please, include the gridlines in the time series and include the number of the month for each year (or the name of the seasons). Fig 5a)

Reply: The difference is the representation in function of latitude in

Figure 3 and of distance for Figure 5a. Figure 5a is mainly there to support the methodology description.The gridlines and number of months have been included.

- ○ Fig. 5-caption: Please, include the meaning of the vertical dashed lines, horizontal grid lines and arrows shown in the a & b panels, as mentioned in L339-L341.
  Reply: Right it has been done

- ○ Fig. 6 – caption. Please, replace "The blue envelope and green bars represent the standard deviation at each point" with "The blue envelope and green bars represent the standard deviation at each point for the model and the satellite data, respectively".
  Reply: it has been done

- ○ Fig. 7 – caption. Please, replace "Time space diagrams" with "Hovmöller diagrams"
  Reply: it has been done

- ○ Fig 8 – 10. Please, consider to plot the altimetry distributions bars with light blue color for improving the visualization in the overlapping area.

  Reply: it has been done

Minor corrections/suggestions

These minor revisions, listed below, will hopefully improve the quality of the manuscript before consideration of publication.

- ● Paper needs work in unifying the text (e.g. HF radars, radars; NC characteristics, diagnostics). As different types of radar technologies are mentioned in the manuscript, please, clearly specify if radars are HF, SAR, etc.
  Reply: We have checked the whole text to harmonize it.
- ● L14. Add 'surface' before 'ocean circulation'.
  Reply: it has been done
- ● L19. Add 'surface' before 'coastal circulation'
  Reply: Done
- ● L20-L27. I would suggest a slight rewording for clarity 'In this methodological study we assess the ability of three altimeter missions (i.e. JASON2, SARAL/AltiKa, Sentinel-3A) data to capture the Northern Current sea level signature in the coastal ocean, using a previously validated high-regional model as a reference. The impact of the recent progress made on both altimetry sensors and data processing on the observation of the NC is also analyzed'. Or something like that.

  Reply: Thanks for the rewording. We replaced the paragraph by "In this methodological study we assess the ability of three altimeter missions (i.e. JASON2, SARAL/AltiKa, Sentinel-3A) data to capture the Northern Current sea level signature in the coastal North Western Mediterranean Sea, using a previously validated high-regional model as a reference. The impact of the recent progress made on both altimetry sensors and data processing on the observation of the NC is also analyzed."

- ● L47. Remove "…" before the parenthesis.
  Reply: Done
- ● L55. Add 'LRM (low-resolution mode)' after 'the Ka-band'.
  Reply: Done

- L90-L99. Rewrite the paragraph to make the main aim of the study and the successive steps given to apply the methodology clearer.
  Reply: The paragraph has been rewritten into "In this paper, we propose a different strategy based on a high resolution numerical model. Our purpose is to assess the ability of satellite altimetry, using three different technologies, to observe a particular coastal dynamical structure new technologies. Using a high resolution model may overcome the issue of colocation between in situ and altimetry, but given the essential condition that the physical process studied must be correctly represented by the model. Our methodology relies first on a careful model validation step in the study region. Then the model is considered as a reference. Our approach will consist in using the model to quantify the SSH signature of an identified physical process along a particular satellite track. In a second time, the model solution will be compared with the SSH signature captured in the altimetry dataset along the considered tracks and the resulting geostrophic currents."
- L102. Remove 'again'.
  Reply: We removed the word "again"
- L107. Include a reference describing the MOOSE. Tintoré et al., 2019 could perhaps be included.
  Reply: Thanks for the suggestion. We have added the reference Tintoré et al., 2019.
- L116. Please double check the transport values. A maximal transport of 1.6 Sv in December is mentioned by Alberola et al., (1995).
  Reply: We have reworded accordingly.
- L120. Please, double check the NC width information. Following the reference given, the NC width is > 30 km with a well-defined episode of narrowing (< 20 km) from late January to mid-March.
  Reply: We have completed the NC width information.
- L135. Please, consider to include also in the 'area south of Toulon' (i.e. HF radar coverage area).
  Reply: We have included this area.
- L155. Add 'HF' before 'radars'.
  Reply: We added "HF" before "radars.
- L156. Add 'to get rid of high-frequency processes not compatible with the hypothesis of geostrophy' after 'oscillations'.
  Reply: Thanks for this precision. We added the sentence after "oscillations"
- L179. Remove '(Synthetic Aperture Radar)' after 'SAR' since it has already been mentioned above.
  Reply: It has been removed
- L193. Replace 'X-TRACK' with 'X-TRACK, Along track Sea Level Anomalies (Version 1.02 - 2017 – DOI: 10.6096/CTOH_X-TRACK_2017_02).
  Reply: We have replaced 'X-TRACK' with 'X-TRACK, Along track Sea Level Anomalies (Version 1.02 - 2017 – DOI: 10.6096/CTOH_X-TRACK_2017_02)'
- L202. Replace '(MDT, Rio et al., 2014)' with 'SMDT-MED-2014, developed by Rio et al., 2014' and further explain that the sum of the MDT and the SLA produces the absolute dynamic topography (ADT), from which the absolute geostrophic velocity is derived using the geostrophic equation (Eq. 1).
  Reply: We changed the paragraph 'From SLA data, the across-track geostrophic current (u) can be inferred through the geostrophic equation (Eq. 1) after adding a regional Mean Dynamic Topography (MDT, Rio et al., 2014).' to 'To obtain Absolute Dynamic Topography (ADT), the SLA data are added to a regional Mean Dynamic Topography (SMDT-MED-2014, developed by Rio et

al., 2014). Then the absolute across-track geostrophic velocity (*u*) is derived from the geostrophic equation (Eq 1).'

- L207-210. I would suggest a slight rewording for clarity 'Both, unfiltered and filtered 1Hz SLA data have been considered for the computation of geostrophic velocities in sections 4.1 and 4.2, respectively'.
  Reply: We have introduced the rewording in the text.
- L219. Replace ';it is described' with ', as described'.
  Reply: Done
- L228. Citation to the model ECMWF (European Centre for Medium-Range Weather Forecasts) is missing. Please, add the corresponding reference.
  Reply: We have added "based on the high resolution 10-day forecast (HRES product)"
- L234. Replace 'Simulation validation' with 'SYMPHONIE model assessment'.
  Reply: Done
- L238-239. I would suggest a light rewording for clarity 'The model performance to represent the NC signal in the velocity field is assessed quantitatively (i.e. time-average and standard deviation) and the NC variability is evaluated qualitatively (Hovmöller diagrams)'.
  Reply: We have reworded the paragraph into "The model performance to represent the NC velocity field in the study area is assessed quantitatively in terms of statistics (time-average and standard deviation) and qualitatively in terms of complete range of variability (Hovmöller diagrams)"
- L251. Add 'HF' before 'radars'.
  Reply: Done
- L252. Remove blank space before 'NC core'.
  Reply: Done
- L260. Replace 'time space diagrams' with 'Hovmöller diagrams'.
  Reply: Done
- L273. Replace 'more differences' with 'higher differences'.
  Reply: Done
- L273. Add 'HF' before 'radars'.
  Reply: Done
- L279. Replace 'time space diagrams' with 'Hovmöller diagrams'.
  Reply: Done
- L280-L281. Please, double check. This consideration ('…misplaced current in the model…') seems to be inconsistent with the sentence above (L277: 'The NC is thus well located in the simulation…').
  Reply: Thanks for this comment. The NC is well located in the simulation in relation to the coast but incorrectly placed in time. We have clarified accordingly.
- L294-L297. Rearrange the paragraph considering the suggested changes in Fig. 4 (above).
  Reply: The paragraph has been removed in the final version.
- L308. Replace 'weaker' with 'lower'.
  Reply: Done
- L310. Replace 'don't with 'do not'.
  Reply: Done
- L321. Please, consider to include also in the 'area south of Toulon' (i.e. HF radar coverage area).
  Reply: Done
- L333. Please, include reference(s) for the selected criteria used to define the width of the NC.

Reply: Done
- L335. Replace 'defining' with 'considered as'.
Reply: Done
- L339-L341. Move this information to the caption of the Fig.5, as requested above.
Reply: We have added the information in Fig.5 caption.
- L355. Add some references after 'Previous studies', as Alberola et al., 1995.
Reply: Done
- L366. Replace 'rms' with 'RMS'.
Reply: Done
- L369-370. Rewrite this paragraph, please.
Reply: This paragraph has been rewritten in "If we consider the global RMS mean error level for the altimetry missions which is 2.23/1.66/1.12 cm for Jason-2/SARAL/S3A, respectively (Vergara et al., 2019), the NC signature on SSH corresponds to greater values and thus might be observable. But its width is generally below the scales resolved. Indeed Jason satellites can capture offshore dynamical signals down to ~70 km wavelength and SARAL/AltiKa and Sentinel-3 down to 35-50 km (Raynal et al., 2017). We also know that the observation of near-shore SSH estimates is a technical challenge for altimetry (Vignudelli et al., 2011). In the next section, using the model as the reference, we analyze which part of the NC SSH and current signals are really sampled by altimetry data."
- L389. Replace 'using the geostrophic equation' with 'using Eq. 1'.
Reply: Done
- L395-L401. I would suggest a light rewording for clarity 'For Jason-2 and SARAL missions, periods were selected based on the joint availability of both, observations and model outcomes (see in Table 2). For Sentinel-3, the matching period was very short, so thus the full data availability periods for observations and model were considered.'.
Reply: It has been reworded.
- L410. Please include '(see Table 1)' if the previous suggestions for Table 1 have been considered by the authors.
Reply: Done
- L424-426. Please, notice that this result (i.e. lower current variability in the model near the coast) is similar to the one for section 2.3 (L255-257), when the model was compared vs. HF radar.
Reply: This has been added in the text.
- L461. Replace 'Note also that the NC is better (almost entirely) resolved in Sentinel-3, compared to Jason-2 and SARAL' with 'Note that Sentinel-3 data better matches the model outcomes in two (i.e. NC width and core location) of the three analyzed diagnostics, while SARAL is closer to the model estimation of the SSH drop'.
Reply: It has been replaced
- L466. Remove 'an operation that strongly…'.
Reply: It has been removed
- L471. Replace 'time space diagrams' with 'Hovmöller diagrams'.
Reply: Done
- L473. Please, specify the reason for clarity.
Reply: We have specified the reason.
- L485. Remove blank space before 'the model current…'.
Reply: Done
- L511. Unit split across lines.

Reply: This has been corrected
- L533. Add 'glider' after (in situ).
  Reply: 'glider' has been added after 'in situ'
- L540. Please, double check the values (same comment as in the abstract).
  Reply: We have checked the values.
- L542. This sentence ('In winter...NV width also tends to diminish') is not consistent with the results found in this work (see L355).
  Reply: This sentence has been removed.
- L543-L545 and L565-L569. please move these paragraphs to the respective sections where the results are shown and discussed.
  Reply: The first sentence has been removed. The second one has been moved to the end of section 4.2..
- L546. 'Larger or lower' the temporal resolution?
  Reply: This sentence has been reworded.
- L550.Please, double check the values. As mentioned in section 4.1, Jason 2 and SARAL  1 Hz-data stop at 8  and 16 km to the coast, respectively.
  Reply: Right, we have changed this sentence accordingly. Now: "We then analyze the NC signature in altimetry data in comparison to the model reference. Jason 2 and SARAL 1-Hz data stop at 8 and 16 km from the coast, respectively, sometimes preventing observation of the whole NC."
- L553. Replace 'too high' with 'overestimated'.
  Reply: Done
- L561. Please, double check this conclusion. As mentioned in section 4.1, Sentinel-3 seems to be the mission that captures the NC almost entirely (L447).

  Reply: This sentence has been reworded.

---

## Author Comment (AC3)

**Assessment of the observability of coastal currents in LRM and SAR altimetry observations: a north-western Mediterranean Sea case study**

**General comments:**

The paper deals with an original method to evaluate the ability of altimetry to catch the main features of a coastal slope current having a space scale in the range of few ten kilometres. A validated numerical modelling is used to fill the gap in terms of space and time co-localisation between in situ measurement and altimetric tracks. The method is innovative and worthy.

Page 3, observability is defined as the condition that the observed processes have a sea level signature and spatial-temporal scales larger than the altimeter resolution. For the north-western Mediterranean Sea, the objective is therefore to check whether altimetry is capable of capturing the observed behaviour of the North Current (mean characteristics, trend, seasonal variability, higher frequency variability of the order of 2 to 15 days, etc.).

Unfortunately, the paper fails to reach this goal and suffer of weaknesses and is not suitable for pulication as is. It should be rejected or strongly revised before any publication.

1) The paper only considers the mean characteristics of the North Current (NC), and the variability is discussed only in terms of standard deviation without distinguishing between measurement noise and physical variability. Of course, the mesoscale perturbations of the NC remain out of the scope as the revisiting period of satellite is too coarse, but one expected at least a discussion about the observability of the seasonal variability. The figure 5.c suggest strongly a comparison with altimetric observations that is not achieved.

Reply: Thanks for this comment. In order to discuss this point we have added hovmöller diagrams at the end of the paper representing the currents after the optimal filtering we found thanks to the histograms. We have commented these figures in the text by focusing on the seasonal variability shown by the hovmöller diagrams but also on some strong events that occurred over the altimetry periods. For Jason 2 and SARAL we have discussed the similarity with the model reference but not for Sentinel-3 as the period is not the same. We have focused on the current amplitude, its width and the location of the NC core.

2) You use the MDT of Rio et al (2014) which reproduces the NC mean slope rather well. One would thus like to know the respective contribution of the SLA and of the MTD to the altimetry derived characteristics of the NC. That is, what is the benefit of adding an SLA to derive the mean characteristics over the concerned time period? Using the longest common available window between altimetric data and numerical modelling prevent any investigation in term of variability and lead only to global statistics.

Reply: We thank the reviewer for this comment. The Mediterranean Sea benefits from a

good quality MDT. However it is not the case elsewhere. Adding the SLA enables to investigate the variability, and/or the mean over another period than the one used to compute the MDT, along the tracks. The respective contribution of SLA and MDT is especially visible when deriving the currents as we represent the mean of individual along-track velocity profiles. To better indicate this contribution we have added the current derived from the MDT in black on Figure 6. We have also developed the text to answer this question by adding the following paragraph: "As we focus on the mean SSH over a long period the results are close to the MDT along the section. However the contribution of the SLA is given by the variability indicated by the error bars. The current obtained from the average of individual current profiles compared to the one derived from the MDT also shows the impact of focusing on the SLA."

3) I don't agree with your chapter 4.2 and associated figures 8,9,10 (maybe I don't understand correctly your methodology?). I guess (it is not written explicitly) you filtered -spatially- only the SLAs before adding the MDT and then derive the current trough geostrophy using your relation (1). Using a low-pass filter with a cut-off wavelength of 60, 50 or even 30 km will remove almost all traces of the Northern Current since it has a horizontal cross-sectional scale of about 20 km. Consequently, the figures 8,9,10 mainly show the distribution of the current derived from the MDT signal appearing when removing progressively the part of NC in the SLA signal. The MDT is more or less in aggreement with the numerical modelling.

Reply: The reviewer is right. We have filtered the SLAs before adding the MDT and then derived the current. We have added this explicitly in the text in section 2.1.c: "Before adding the MDT and computing current estimates, the SLA may be filtered in the along track direction in order to remove the remaining altimetry noise". We have also rewritten the methodology used in section 4.2 in order to clarify it and better explain our objectives.

The histograms represent the variability of the current obtained after the filtering of the data. The current derived from the MDT only does not reproduce the model distribution. Please find below the histogram of the model velocities along the Jason 2 track with the MDT velocities superimposed. To compare both products we repeated these values as many times as the satellite passed over the track.

[Figure]

*Figure 1: Distribution of the current values along the Jason 2 track 222 over the first 60 km to the coast for the model (in pink) and the current derived from the MDT (in blue)*

The currents derived from the MDT show a peak of strong values at about -0.3 m/s. This is not the case for the model as it shows the current variability with some values going until -0.6 m/s. To support the assumption that the histograms represent the variability of the Northern Current we have plotted the results obtained for winter and summer months. You can find these figures below. We do not add these results in the paper however we comment on the variability shown by the histograms in the text.

[Figure]

[Figure]

*Figure 2: Distribution of the current values along the Jason 2 track 222 over the first 60 km to the coast for low-passed filtered altimetry data with a cutoff frequency of 60 km (in blue) and the model (in pink) for the winter months (January, February, March)*

*Figure 3: Distribution of the current values along the Jason 2 track 222 over the first 60 km to the coast for low-passed filtered altimetry data with a cutoff frequency of 60 km (in blue) and the model (in pink) for the summer months (July, August, September)*

The methodology used reduces the ambition of the study. The benefit of the different altimetric signal (Jason,Saral,Sentinel) is not fully demonstrated as we don't know the SLA's own contribution to the current mean and as the physical variability is mixed with the noise. May I suggest to do at least seasonal means in order to investigate if the SLA is able to catch the seasonal variation of the NC. It seems possible on longer series.

Reply: We hope that the addition of the figures and text previously described allow to answer this comment. We have discussed the SLA's contribution in section 4.1 and added a paragraph in section 4.2 to investigate the current variability with the hovmöller diagrams. These hovmöller diagrams also enable to visualize to which extent the NC is captured: almost entirely for Sentinel-3 whereas it is not completely resolved for Jason 2. SARAL current is also less noisy compared to the other missions.

**Detailed comments:**

Lines 122-128: You refer to the variability of the NC without indicating a time scale or length. It might be useful for the reader to have this information in relation to the frequency of the model outputs and the satellite repetition. In the literatures, two periods dominate 10-20 days and 2-6 days.

Reply: Thanks for this suggestion. We have added in the text the time scales in that paragraph.

Line 210: You should explain why you are or are not applying filtering.

Reply: We have added "To investigate the data noise issue, both unfiltered and filtered SLA have been considered for the computation of geostrophic velocities in sections 4.1 and 4.2 respectively"

Line 250: amplitude is perhaps not the exact terms as it refers here to the mean value of the NC core velocity.

Reply: We have changed the term "maximum NC amplitude" to "mean NC core velocity"

Figure 3: A suggestion, a white centred palette would be more appropriate to illustrate the velocity differences.

Reply: it has been done

Line 280: "They are associated with a misplaced current in the model rather than with incorrect current values". ? You mean probably "incorrect current intensity" or more precisely "incorrect current maximum".

Reply: We have changed "incorrect current values" to "incorrect current maxima"

Line 290: "The irregular temporal sampling of the gliders also contributes to these larger model-data differences, compared to the HF radars results. Indeed, a deeper analysis shows that the same features may occur in the simulation and in the observations, but shifted by one or two days (not shown)." I don't understand why time lags in signal induce more differences for irregular sampling than for regular one. Figure 3c exhibits also strong difference for radar comparison with the HF.

Reply: Here we wanted to highlight that a regular and high frequency sampling as the one of the HF radars (every day) enables to find the same signal but with an offset of 2-3 days for example. The structures are just lagged in time in the Hövmoller diagrams which explain the differences between the model and the radars in Figure 3c while qualitatively the diagrams seem really close. However as there are gaps in the gliders sampling, if a structure is offset it will not appear in the model. To be clearer in the text we have reformulated into "The irregular temporal sampling of the gliders also contributes to these larger qualitative model-data differences, compared to the HF radars results. Indeed, a deeper analysis shows that the same features may occur in the simulation and in the observations, but shifted by one or two days (not shown). Thus they are represented in the HF radars Hovmöller diagram but may correspond to gaps in the glider diagram."

Figure 4: In my opinion, figure 4 is not really useful for your demonstration and the associated paragraph (line 286-314) is confusing.

Reply: We have removed Figure 4 and the associated paragraph.

Figure 5: To support the corresponding text, dx, |u|max, |u|max/2 must be quoted in figures 5a and b, otherwise these figures are not helpful.

Reply: Thanks for this suggestion, we have included dx, |u|max, |u|max/2 in the figure.

Line 466: The increase in noise due to spatial filtering does not seem to be addressed in section 4.1.

Reply: The sentence was confusing. It has been changed to "In practice, users systematically apply a spatial filter to altimetry SLA data before geostrophic current derivation in order to remove the measurement noise observed in section 4.1"

Typo:

Line 97: the3 -> the

Reply: It has been corrected

The name of the journal is missing for several references.
Reply: It has been corrected